# Foundation Model Informed Acquisition Functions for Molecular Discovery

## Abstract

Bayesian Optimization (BO) is an effective approach for accelerating molecular discovery by iteratively learning probabilistic surrogates of the mapping from molecules to their properties and optimizing the corresponding acquisition functions to select promising candidates. However, traditional BO struggles in low-data regimes with limited prior knowledge and vast search spaces. Large language models (LLMs) and chemistry foundation models provide rich priors to enhance BO, but their use is hindered by high-dimensional representations, costly in-context learning, and the computational burden of deep Bayesian surrogates. To address these challenges, we propose a likelihood-free BO framework that bypasses explicit surrogate modeling and directly leverages priors from general LLMs and chemistry foundation models to inform acquisition functions. Our method learns a tree-structured partition of the molecular search space with local acquisition functions, enabling efficient candidate selection via Monte Carlo Tree Search. Additionally, coarse-grained LLM-based clustering improves scalability by restricting acquisition evaluations to clusters with statistically higher property values. Extensive experiments and ablations demonstrate substantial improvements in scalability, robustness, and sample efficiency for LLM-guided BO in molecular discovery.

## 1. Introduction

Discovering molecules with desirable properties is crucial for drug design, materials science, and chemical engineering. Given the vast chemical space (Restrepo, 2022), exhaustive evaluation is infeasible, as Density Functional Theory (DFT) simulations (Parr, 1989) are computationally expensive and experimental evaluations are laborious and time-consuming. Bayesian Optimization (BO, Frazier (2018); Garnett (2023)) aims to minimize costly evaluations and accelerate discovery by using Acquisition Functions (AFs), expressed as the expected utility under a surrogate model (*e.g.*, Gaussian Processes (GPs) and Bayesian Neural Networks (BNNs)) to guide the search toward promising candidates, balancing exploration of uncertain regions with exploitation of high observed-value regions.

However, the high cost of evaluations limits the number of initial points available to seed BO with informative acquisition function priors (typically ∼10 samples used in previous studies (Xie et al., 2025; Kristiadi et al., 2024)), further constraining its performance. Recent approaches incorporate LLM priors into BNNs (Kristiadi et al., 2024) with fixed features, Parameter-Efficient Fine-Tuning (PEFT; e.g., Low-Rank Adaptation (LoRA (Hu et al., 2021))), or In-Context Learning (ICL; (Ramos et al., 2023)), but they remain limited by scalability, cost, and computational challenges due to the vast discrete candidate space.

To address these challenges, we propose a principled foundation-model-informed Bayesian optimization method, `LLMAT(LLM-guided Acquisition Tree)`, as illustrated in Fig. 1. LLMAT **(a)** avoids the costly learning of a Bayesian surrogate model by performing likelihood-free AF estimation; **(b)** leverages rich prior knowledge from both general-purpose LLMs and domain-specific foundation models to inform AFs; and **(c)** partitions the molecular candidate space into a tree structure, learning a local AF at each node and enabling efficient candidate selection via Monte Carlo Tree Search (MCTS) at each BO iteration, even in the presence of high-dimensional LLM-derived features.

Concretely, we directly model local AFs via density ratio estimation, obtained by optimizing a weighted binary classification objective at each tree node. These classifiers determine both the tree partitions and the corresponding local AFs. By meta-learning the shared LoRA weights and the initialization of the root-node classifier, we improve the stability of both the partitioning process and PEFT updates under low-data regimes. Further ablation studies (Fig. 9) validate the effectiveness of these two modules.

To better inform AFs with prior from large foundation models, we first observed that, although general LLMs are pri-

---
[1]Anonymous Institution, Anonymous City, Anonymous Region, Anonymous Country. Correspondence to: Anonymous Author <anon.email@domain.com>.

Preliminary work. Under review by the International Conference on Machine Learning (ICML). Do not distribute.

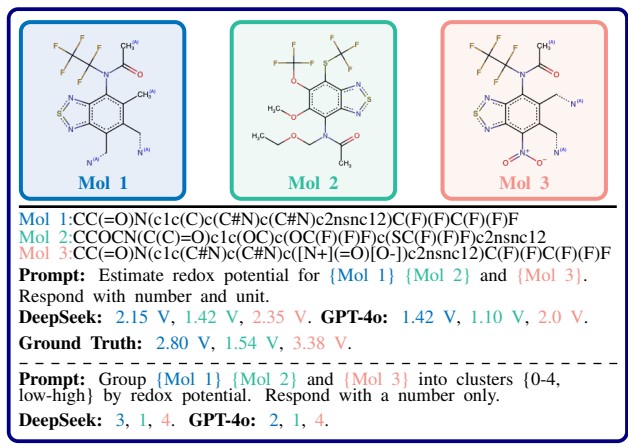

*Figure 1.* Overview of LLMAT: Binary classifiers on the LLM features recursively partition the candidate space to a tree structure and local AFs are learned from observations in each partition $\Omega_k \cap \mathbf{D}_t$. Promising partition $\Omega_{\text{selected}}$ is selected via UCB. Unseen candidates are clustered via LLM prompts, and promising clusters $\Omega_{\text{cluster}}$ are selected via statistical tests. Within $\Omega_{\text{cluster}} \cap \Omega_{\text{selected}}$, the property of the candidate with the highest local AF value is evaluated and added to the dataset.

marily trained on natural language and cannot predict precise numerical property values without in-context learning (ICL), they can capture coarse-grained information, such as whether a molecule's property is relatively high or low. As shown in Fig. 2, while LLM-predicted redox potentials are not numerically accurate, the relative ordering of molecules is largely preserved. Building on this observation, we introduce an LLM-based clustering phase that queries a general LLM once to assign cluster labels to the entire candidate set. This cluster information is then used to improve scalability and BO performance by restricting AF evaluations to clusters with statistically higher property values.

To summarize, the main contributions of our paper are:

**A novel molecular discovery framework** that informs AFs using both general LLMs and specialized foundation models, mitigating key challenges in prior BO methods in a systematic way. Extensive experiments on six real-world chemistry datasets demonstrate that it outperforms baselines in most cases with improved scalability, benefiting from the prior knowledge in general LLMs, chemistry foundation models, and the improved algorithm design.

**A principled discrete BO method** that uses shared binary classifiers to construct a tree-structured partition of the molecular search space and learn local AFs at each node at the same time, enabling efficient candidate selection and AF estimation via Monte Carlo Tree Search. To improve stability in low-data regimes, we further introduce a meta-learning strategy for training these classifiers, resulting in more reliable and generalizable AFs across partitions.

**A generic LLM-guided pre-clustering strategy** that estimates AF values only for candidates within statistically selected clusters. This plug-in, method-agnostic module reduces computational cost, especially for methods using PEFT, improving scalability for large candidate sets. It provides a practical way to incorporate prior from general LLMs into algorithm design with reasonable API cost.

*Figure 2.* General LLMs provide coarse property ranking of redox potentials rather than accurate numeric value for the three given molecules; responses are color-matched.

## 2. Related Work

Due to space limitations, a more detailed discussion of related works is provided in Appendix B.

**LLMs for Molecule Discovery.** Recent studies have explored the potential of LLMs in a variety of chemistry-related tasks, including molecular property prediction (Lu & Zhang, 2022; Christofidellis et al., 2023; Guo et al., 2023; Jablonka et al., 2023), property-related molecule optimization (Ramos et al., 2023; Kristiadi et al., 2024), and molecular generation with desired properties (Liu et al., 2024a; Flam-Shepherd & Aspuru-Guzik, 2023; Wang et al., 2024; Jablonka et al., 2024). However, many of these LLM-based approaches (Ramos et al., 2023; Guo et al., 2023) rely heavily on in-context learning (ICL) and prompt engineering. This dependency poses challenges for tasks involving strict numerical objectives, where LLMs often fall short in satisfying precise quantitative constraints or optimizing for specific target values (AI4Science & Quantum, 2023). To address this limitation, recent efforts have augmented LLMs with

optimization frameworks such as Bayesian Optimization (BO) or Evolutionary Algorithms (EA). For example, Kristiadi et al. (2024) integrate Laplace approximations with Bayesian neural networks for property prediction, while Wang et al. (2024) combine EA to generate and search molecules with desired property.

**LLMs for Bayes Optimization over Molecules.** General LLMs and chemistry foundation models capture rich priors from text and molecular datasets, offering the potential to guide BO in low-data regimes. Recent studies have explored this potential through various adaptations. For example, Ramos et al. (2023) implemented BO via ICL by adaptively prompting general-purpose LLMs like GPT-4, which is effective but costly due to accumulated API queries and limited by prompt length. Alternatively, Kristiadi et al. (2024) used LLMs for fixed-feature extraction and PEFT (Hu et al., 2021; Li & Liang, 2021) in the BO loop, requiring expensive Laplace approximations of BNNs across the full candidate set and imposing high computational and memory demands. Moreover, relying on a single surrogate trained on limited data can be suboptimal, especially for high-dimensional LLM embeddings (Wang et al., 2020). In contrast, our method directly leverages these priors to guide acquisition functions without costly surrogate learning and incorporates LLM-based clustering to restrict evaluations to promising regions, enabling scalable and data-efficient BO in large, high-dimensional molecular spaces.

## 3. Background

### 3.1. Bayesian Optimization

Bayesian Optimization (BO) aims to effectively maximize some unknown function $f : \mathcal{X} \to \mathcal{Y}$ over the candidate space $\mathcal{X}$, *i.e.*, find $x^* = \arg\max_{x \in \mathcal{X}} f(x)$ [1], given a dataset $\mathbf{D}_t = \{x_i, y_i\}_{i=1}^t$ of $t$ observations, where $x_i$ is a molecule, $y_i = f(x_i) + \epsilon, \forall i \in [t]$ is a property value, and $\epsilon$ is a noise conventionally assumed to be a Gaussian with $\epsilon \sim \mathcal{N}(0, \sigma^2)$. Due to the intractability of $f$, a probabilistic surrogate model $\hat{f}$ learned from $\mathbf{D}_t$ is often used to approximate $f$ with consideration of epistemic uncertainty, reflected in the variance of the posterior $p(\hat{f}|\mathbf{D}_t)$. Hence, for any unobserved data $(x, y)$, we have the predictive posterior $p(y|x, \mathbf{D}_t) = \int p(y|\hat{f}(x))p(\hat{f}(x)|\mathbf{D}_t)d\hat{f}(x)$.

**Acquisition Function and Expected Utility.** To address problems in the sequential decision-making setting, BO methods typically select the next query $x_{t+1}$ by maximizing an acquisition function $\alpha : \mathcal{X} \to \mathbb{R}$, defined as:

$$\alpha(x; \mathbf{D}_t, \tau) := \mathbb{E}_{y \sim p(y|x, \mathbf{D}_t)}[u(y; \tau)], \quad (1)$$

where $x_{t+1} = \arg\max_{x \in \mathcal{X}} \alpha(x; \mathbf{D}_t, \tau)$, $u(y; \tau)$ is a chosen utility function and $\tau$ is a threshold that measures the

[1]It can be extended to minimization without loss of generality.

utility of $y$. The selection of $\tau$ controls the exploration-exploitation trade-off, and sometimes is set as $\tau = \max_t y_t$, the best observed value so far. Common choices of the utility function $u(y; \tau)$ satisfying Eq.(1) include: $u^{EI}(y; \tau) = \max(y - \tau, 0)$ for Expected Improvement (EI, Mockus et al. (1978)) that measures how much $y$ exceeds the threshold $\tau$; $u^{PI}(y; \tau) = \mathbf{1}(y - \tau > 0)$, indicating whether $y$ exceeds $\tau$ for Probability of Improvement (PI, Kushner (1964)); Many other AFs can be expressed in this form with a more complex utility function, such as Upper Confidence Bound (UCB, Srinivas et al. (2009)), Entropy Search (ES, (Hennig & Schuler, 2012)), Knowledge Gradient (KG, Frazier et al. (2008)), Thompson Sampling (TS, Thompson (1933)).

**Direct Estimation of Acquisition Functions.** Typical BO instantiations can be characterized as *indirect*: they first approximate the predictive posterior $p(y \mid x, \mathbf{D}_t)$ via a surrogate model based on Gaussian Processes (GPs) or Bayesian Neural Networks (BNNs), and then compute the acquisition functions via Eq. (1) for a given utility function. We provide an introduction to this in Appendix A.1. However, these methods often face scalability issues due to the high computational cost when combining large, deep models, which limits their flexibility. Our proposed method on the other hand directly models AFs, forgoing the need for surrogate model. Let $l(x) = p(x|y \le \tau, \mathbf{D}_t)$ and $g(x) = p(x|y > \tau, \mathbf{D}_t)$ be two densities that respectively characterize the data distribution on non-promising and promising region of candidate space. Instead of explicitly modeling the predictive posterior $p(y|x, \mathbf{D}_t)$, Bergstra et al. (2011) and Tiao et al. (2021) directly model the AF as $\gamma$-relative density ratio:

$$r_\gamma(x) := \frac{g(x)}{\gamma g(x) + (1 - \gamma)l(x)} = h_\gamma(r_0(x)), \quad (2)$$

where $h_\gamma : u \to (\gamma + u^{-1}(1 - \gamma))^{-1}, u > 0$ is strictly non-decreasing, $r_0(x) = \frac{g(x)}{l(x)}$ is the ordinary density ratio, and $\gamma = p(y > \tau | \mathbf{D}_t)$ indicates $\tau$ is set as the $\gamma$-th quantile of all observed $y$. This density ratio has been proven by Song et al. (2022) to be equivalent to PI . Tree Parzen Estimator (TPE, (Bergstra et al., 2011)) estimates the density ratio $r_0(x)$ by separately estimating $\ell(x)$ and $g(x)$ with tree variant of Kernel Density Estimation (KDE). BORE (Tiao et al., 2021) avoids such indirect estimation by formulating the problem as a binary classification and showing the classifier $\pi_\theta(x) \approx p(c = 1|x, \mathbf{D}_t) = \gamma r_\gamma(x)$, where $c = \mathbf{1}(y > \tau)$ is the class label and $p(c = 1|\mathbf{D}_t) = p(y > \tau|\mathbf{D}_t) = \gamma$. Likelihood-free Bayes Optimization (LFBO, Song et al. (2022)) adopts variational $\phi-$divergence estimation to directly estimate expected utility AFs for complex $p(y|x, \mathbf{D}_t)$ and $u(y; \tau)$, not limited to PI. Then, the likelihood-free AF is defined as:

$$\alpha(x; \mathbf{D}_t, \tau) := \pi_{\theta^*}(x)/(1 - \pi_{\theta^*}(x)), \quad (3)$$

where $\pi_\theta(x) := p_\theta(c = 1|x, \mathbf{D}_t)$, $\theta^* = \arg\min_\theta \mathcal{L}_{\mathbf{D}_t}^\tau(\theta)$,

$$\mathcal{L}_{\mathbf{D}_t}^\tau(\theta) := \mathbb{E}_{(x,y) \sim \mathbf{D}_t}[-u(y; \tau)\log \pi_\theta(x) - \log(1 - \pi_\theta(x))].$$

### 3.2. BO for Molecular Discovery

In molecular discovery, the goal is to identify a novel molecule $x \in \mathcal{X}$ with desirable properties $y \in \mathcal{Y} \subseteq \mathbb{R}$, often requiring exploration of a **vast** and **discrete** search space $\mathcal{X}$ (estimated to contain over $10^{100}$ unique molecules). Exhaustive property evaluation is infeasible and costly, e.g., when relying on Density Functional Theory (DFT). In practice, we often restrict the search to a smaller candidate subset $\Omega \subset \mathcal{X}$. At each round $t + 1$, given prior observations $\mathbf{D}_t = \{x_i, y_i\}_{i=n+1}^{n+t} \cup \mathbf{D}_0$ with $n$ initial points $\mathbf{D}_0 = \{x_j, y_j\}_{j=1}^{n}$, a new candidate $x_{t+1}$ is selected from $\Omega \setminus \mathbf{D}_t := \{x_i \in \Omega \mid x_i \notin \mathrm{supp}_X(\mathbf{D}_t)\}$ for evaluation, aiming to find the optimal molecule $x^*$ in as few rounds as possible. This naturally fits the sequential BO framework.

To input molecules into machine learning models, common representations (Griffiths et al., 2023; Janakarajan et al., 2024) include: (1) text-based formats such as SMILES (Weininger, 1988), SELFIES (Krenn et al., 2020), and IUPAC names; (2) feature-based fingerprints such as Morgan (Morgan, 1965) and ECFPs (Rogers & Hahn, 2010); and (3) graph-based encodings (Duvenaud et al., 2015). In this work, we focus on text-based representations, which can be directly generated by modern NLP models, including general-purpose LLMs, transformer-based chemical foundation models, and domain-specific LLMs specialized for chemical data.

## 4. Methodology

As discussed previously, the early rounds of BO with limited observations, the high dimensionality of molecular features, and the vast discrete candidate space pose major challenges for efficient discovery. We address these by informing foundation models to achieve: **(1) Refined AF optimization.** Local AFs are learned on top of foundation model feature extractors while building the tree partition, combined with meta-learning of the root node initialization to mitigate overfitting for each sub-node. The next candidate is selected by comparing AF values within a small set of molecules in the most promising leaf node. **(2) Improved computational efficiency.** Evaluating AFs across the full candidate space is costly, especially with PEFT, so we employ LLM-based clustering and statistical testing to prune the search space before AF estimation. We refer to our method as *LLM-guided Acquisition Tree* (LLMAT), which integrates LLM-based clustering, foundation model representations, and tree-structured acquisition function learning for efficient molecular discovery. An overview is shown in Fig. 1.

### 4.1. Meta-Learning Candidate Partitions and Local AFs

Now we introduce how to train shared classifiers for candidate partitioning and local AFs during tree construction.

**Recursive Candidate Space Partitioning.** At any iteration $t$, we have the currently observed dataset $\mathbf{D}_t = \{x_i, y_i\}_{i=1}^{n+t}$ defined in Sec. 3.1. As shown in Fig. 1, we use Monte Carlo Tree Search (MCTS) to hierarchically partition the candidate set $\Omega$, where each node in the tree corresponds to a subset of $\Omega$. The tree construction begins at the root node (node 0) with $\Omega_0 = \Omega$ being the entire candidate space.

Each node $\Omega_k$ is recursively bifurcated into two subsets using a binary classifier $\pi_{\theta_k}$. Let $\Omega_{2k+1} := \{x \in \Omega_k \mid \pi_{\theta_k}(x) > 0.5\}$ and $\Omega_{2k+2} := \{x \in \Omega_k \mid \pi_{\theta_k}(x) \leq 0.5\}$ be the left and right children respectively, representing the more promising (i.e., more likely to contain high-property candidates) and less promising regions. The tree expansion continues until a predefined maximum depth $L$ or the least number of samples in each node is reached, yielding a total maximum of $2^{L+1} - 1$ nodes. To train the binary classifier at each node $\Omega_k$, we use the subset of observed samples within that node, i.e., $\mathbf{D}_t \cap \Omega_k := \{(x_i, y_i) \in \mathbf{D}_t \mid x_i \in \Omega_k\}$, where their class labels can be obtained by directly thresholding $y$ as we described in the following. For unobserved samples in $\Omega_k$, the learned classifier $\pi_{\theta_k}$ assigns them to the left or right child. When estimating the AFs, rather than classifying all samples across the whole tree, we only route the search from the root to the most promising leaf, using classifiers at each node along that path.

---

**Algorithm 1** Meta-learning Tree Partitions and local AFs

---

1: **Input:** Observed dataset $\mathbf{D}_t$, quantile $\gamma$, candidate set $\Omega$, tree depth $L$, classification head $\theta$, PEFT parameters $w$, learning rate $\eta$, SGD steps $K$, finetuning flag $\mathbb{F}$, minimal leaf sample size $m$.
2: Nodes $\mathcal{N} = \{\}$;
3: $\Omega_0 = \Omega, \mathbf{D}_t \cap \Omega_0 = \mathbf{D}_t, \mathbf{D}_t \cap \Omega_k = \emptyset, \forall k > 0$;
4: **for** $k = 0$ to $2^{L+1} - 2$ **do**
5:     **if** $|\Omega_k \cap \mathbf{D}_t| < m$ or $\Omega_k$ not splitable **then**
6:         continue;
7:     **end if**
8:     $\tau_k = \Phi_k^{-1}(\gamma)$; Fix $w$, update $\theta_k = \mathrm{SGD}(\mathcal{L}_{\mathbf{D}_t \cap \Omega_k}^{\tau_k}(\theta, w), K)$;
9:     $\theta \leftarrow \theta + \eta(\theta_k - \theta), \mathcal{N} \leftarrow \mathcal{N} \cup \{k\}$;
10:     **for** $(x, y)$ in $\mathbf{D}_t \cap \Omega_k$ **do**
11:         **if** $\pi_{\theta_k}(x) > 0.5$ **then**
12:             $\mathbf{D}_t \cap \Omega_{2k+1} \leftarrow (\mathbf{D}_t \cap \Omega_{2k+1}) \cup \{(x, y)\}$;
13:         **else**
14:             $\mathbf{D}_t \cap \Omega_{2k+2} \leftarrow (\mathbf{D}_t \cap \Omega_{2k+2}) \cup \{(x, y)\}$;
15:         **end if**
16:     **end for**
17: **end for**
18: **if** $\mathbb{F}$ **then**
19:     Fix $\theta$, update $w = \mathrm{PEFT}(\frac{1}{|\mathcal{N}|} \sum_{k \in \mathcal{N}} \mathcal{L}_{\mathbf{D}_t \cap \Omega_k}^{\tau_k}(\theta, w))$;
20: **end if**
21: **Output:** $\pi_{\theta_k}(x), \alpha(x; \Omega_k \cap \mathbf{D}_t, \tau_k), \forall k \in \mathcal{N}$;

---

**LFBO Classifier Shared for Bifurcating.** In LFBO, the AF at each iteration $t$ is estimated directly by learning a (weighted) binary classifier $\pi_\theta(x) = p_\theta(y > \tau | x, \mathbf{D}_t)$, instead of training surrogate models for typical BO algorithms. This classifier can be reused as a natural choice for the aforementioned candidate space partitioning during the construction of the tree. At each node $k$, we set $\tau_k = \Phi_k^{-1}(\gamma)$ under the same $\gamma$, where $\gamma = \Phi_k(\tau_k) :=$

$p(y > \tau_k | \mathbf{D}_t \cap \Omega_k)$. Then, the corresponding binary classifier is $\pi_{\theta_k}(x) = p_{\theta_k}(c = 1 | x, \mathbf{D}_t \cap \Omega_k)$ with $c = \mathbf{1}(y > \tau_k)$, where the binary labels $c$ for training the classifier are assigned by thresholding the $y$ for all training samples in node $k$ instead of using the binary clusters in Wang et al. (2020).

**Meta-learning Shared Binary Classifiers.** Using a shared classification head connected after the transformer feature extractor, we propose a meta-learning approach that works for both fixed features and parameter efficient fine-tuning (PEFT). Since the tree is built recursively, we use the sequential version of Reptile (Nichol & Schulman, 2018), meta-training the classification head. For a maximal $L$-depth Tree, we store one meta-model $\theta$ and $|\mathcal{N}|$ local models $\theta_k, \forall k \in \mathcal{N}$. The detailed algorithm for learning the partitions and AFs at each iteration is presented in Algo. 1.

### 4.2. Candidate Selection on Promising Subsets

With the built tree partition and learned local AFs, we can perform refined AF optimization via both partition and cluster selection to identify promising candidates efficiently.

**Partition Selection.** Given the aforementioned partition rule, a greedy strategy that always selects the left node would exploit the most promising leaf but risks overfitting to suboptimal partitions without sufficient exploration. To balance exploration and exploitation, we select partitions using some score metrics. For example, as in Wang et al. (2020), let $n_k = |\mathbf{D}_t \cap \Omega_k|$ denote the visit count for each node $k$, and let the node value $v_k = \frac{1}{n_k} \sum_{i=1}^{n_k} y_i$ be the average property value of observed samples at node $k$. Then, the Upper Confidence Bound for Trees (UCT) score for a node $k \in \mathcal{N}$ is then defined as: $\mathcal{S}_k^{\text{UCT}} := v_k + 2\lambda\sqrt{2\log(n_p)/n_k}$, where $p = \lceil \frac{k}{2} \rceil - 1$ and $n_p$ is the number of visits at the parent node of $k$-th node and $\lambda$ is a hyper-parameter that controls the exploration-exploitation trade-off. Another possible choice is $\mathcal{S}_k^{\text{Var}} := v_k + 2\lambda\sqrt{\text{var}_k}$, where $\text{var}_k = \frac{1}{n_k} \sum_{i=1}^{n_k} (y_i - v_k)^2$. When $\lambda = 0$, both recover the greedy policy. Finally, we select the child node with the highest score from the root to a leaf. If no candidates reach a leaf node, we use a backtracking approach to select the parent node instead. The selected partition is denoted as $\Omega_{selected}$, as shown in Algo. 2.

**Clustering for Efficient Estimation of AFs.** To reduce the high GPU memory and computation cost of calculating AFs for all candidates at each BO round, we propose a clustering-based approach. Molecular feature representations are precomputed once and grouped into clusters. Candidates within the selected clusters then undergo the partition selection process, and the optimal candidate is determined by comparing AFs only for $\Omega_{\text{cluster}} \cap \Omega_{selected}$.

**(1) Cluster Selection via Statistical Test.** To identify promising clusters for reduced AF estimation, we propose a statistical filtering approach based on observed data $\mathbf{D}_t$.

Specifically, Welch's ANOVA (Appendix A.3) that is robust to heterogeneous variances is first tested. For a given p-value $p$, if significant differences are detected in average property values across clusters, the Games-Howell post-hoc test (Appendix A.4) is applied to exclude outlier clusters using the same $p$. To support this, the BO initialization $\mathbf{D}_0$ is constructed by uniformly sampling from each cluster.

**(2) LLMs for Property-Related Clustering.** Unsupervised clustering methods (e.g., k-means) do not guarantee grouping by property values, limiting the effectiveness of our cluster selection strategy. In contrast, general LLMs (e.g., ChatGPT) can provide property-aware clustering by classifying molecules into high, medium, or low property groups via tailored prompts, as what we used in Appendix E. This leverages the chemical knowledge embedded in their training corpora, yielding clusters aligned with property values. As shown in the experiments, it can enhance BO at a cost that is orders of magnitude lower than the ICL-based approaches reported in Kristiadi et al. (2024), thereby providing a highly cost-effective way to exploit general LLMs.

## 5. Experiments

In this section, we experimentally validate the proposed algorithm across multiple datasets and ablation scenarios, demonstrating its ability to accelerate molecular discovery through superior BO performance and increased computational efficiency. All results are reported with variance over 15 independent runs of different random seeds. Additional details and results are provided in Appendix F and G.

### 5.1. Setting

**Datasets.** We employ the well-known multi-modal Levy function (Levy et al., 2006) to generate a synthetic set to demonstrate the effectiveness of the proposed method in achieving more refined AF estimation within localized sub-regions. For molecular data, following Kristiadi et al. (2024), we initialize BO with $n = 10$ points and and evaluate our models on six benchmark datasets that capture realistic molecular design challenges across diverse domains: (1) minimizing redox potential for redoxmers (1,407 molecules), (2) minimizing solvation energy for flow battery electrolytes (1,407 molecules) (Agarwal et al., 2021), (3) minimizing docking scores of kinase inhibitors (10,449 molecules) for drug discovery (Graff et al., 2021), (4) maximizing fluorescence oscillator strength for laser materials (10,000 molecules) (Strieth-Kalthoff et al., 2024), (5) maximizing power conversion efficiency (PCE) in photovoltaic materials (10,000 molecules) (Lopez et al., 2016), and (6) maximizing $\pi - \pi^*$ transition wavelengths for organic photoswitches (392 molecules) (Griffiths et al., 2022). Together, these tasks cover a broad spectrum of molecular properties, providing a comprehensive testbed for molecular discovery.

For each dataset, the physics-based simulators released by the original authors are used as the ground-truth oracles.

**Fixed Features and Foundation Models.** To assess whether the findings of Kristiadi et al. (2024) that LLMs benefit molecular BO only when trained on domain specific data hold for our algorithm, we benchmark LLMAT against several baselines on using different features: (i) Morgan fingerprints (Morgan, 1965) as a chemistry specific algorithmic representation, (ii) chemistry foundation models including T5-Chem (Christofidellis et al., 2023) and MolFormer (Ross et al., 2022), and (iii) general LLMs such as T5 (Raffel et al., 2023), GPT-2 (Radford et al., 2019), and Llama-2-7b (Touvron et al., 2023). Our comparison includes (1) Bayesian optimization using fixed feature representations extracted from these models, and (2) Parameter-Efficient Fine-Tuning (PEFT) of the above chemistry foundation models.

**Evaluation Metrics.** We report the trajectory of the best observed property value over time and use the **GAP** metric defined in Jiang et al. (2020): $\text{GAP}_t := \frac{y_t^* - y_0}{y^* - y_0}$, which normalizes the progression of the optimal value, where $y^*$ is the global optimum, $y_0$ is the initial optimum, and $y_t^*$ is the best value observed up to iteration $t$. However, in molecular discovery, identifying a single optimal molecule is often not the only goal. In practice, it is equally important to discover a *set of high-performing candidates* that scientists can further evaluate, balance multiple criteria, or analyze for structure-property relationships. For this objective, we additionally report the **average regret**: $\text{Regret}_t := \frac{1}{t} \sum_{i=1}^{t} (y^* - y_i)$, which reflects the overall quality of the molecules discovered so far. When aggregating results across datasets, we normalize the average regrets for comparability.

## 5.2. Performance Analysis

**Effectiveness of Localized Search on Synthetic Data.** Beyond LLM adaptation and its application to molecular discovery, our method contributes a novel discrete BO algorithm: we introduce shared binary classifiers that jointly serve both tree partitioning and localized AF approximation. To illustrate this benefit, we sample 1,000 examples from the Levy-1D function and observe that the proposed algorithm yields a more refined AF estimate at the selected leaf node than at the root node used by vanilla LFBO (Song et al., 2022). As shown in Fig. 3, the leaf node exhibits a narrower confidence region and more accurate estimation around the true optimum, and the refinement becomes more pronounced over BO iterations as additional observations accumulate, see Sec. G.6 in the Appendix.

**Superior Performance of LLMAT on Molecular Data.** In the left panel of Fig. 4, we show that the proposed LLMAT algorithm outperforms standard discrete BO baselines when applied to fixed features extracted from the aforementioned models: Gaussian Processes (GP), the Laplace approxi-

mation (LAPLACE) (Kristiadi et al., 2024), Likelihood-Free BO (LFBO), BORE (Tiao et al., 2021), and random search (RANDOM), as measured by the average GAP metric across all datasets and models. The middle and right plot of Fig. 4 further compares LLMAT against LAPLACE across all datasets to assess iterative PEFT effectiveness with T5-Chem and Molformer backbones. While iterative PEFT enhances BO performance for both approaches, LAPLACE with PEFT still falls short of the performance achieved by our algorithm without PEFT. The radar chart in the left of Fig. 7 measures fine-tuning improvements over fixed features across six metrics demonstrates that our method achieves: (1) faster convergence in early BO rounds, (2) superior final performance, and (3) greater stability across different foundation models. The PEFT improvements are more substantial for Molformer than T5-Chem, reflecting T5-Chem's larger scale and richer domain knowledge.

**Better Exploitation of Priors from Fixed Features.** In Fig. 5, we compare LLMAT (with maximal tree depth $L = 1$ of 3 nodes) against GP, LAPLACE, and RANDOM, using various fixed features mentioned before. We only plot the feature that achieved the best performance for each algorithm; the full plot is deferred to Fig. 11. Fig. 5 shows that LLMAT achieves superior performance compared to all baselines on most datasets, with slightly lower performance than LAPLACE on the Laser dataset. Other baselines exhibit substantial performance drops on several datasets while LLMAT remains consistent. LLMAT, LAPLACE, and GP achieve their best performance on the same feature models for the Redox-mer, Kinase, and Laser datasets (T5-Chem, T5, and MolFormer, respectively.). However, the best feature models differ for other datasets. The strong performance of LLMAT with T5, GPT-2-large, llama2-7b, and Morgan fingerprints suggests that general-purpose LLMs can still provide valuable information. This contrasts with Kristiadi et al. (2024), who argue that LLMs are useful for BO over molecules only when pretrained or finetuned on domain-specific data, highlighting the importance of algorithmic design to fully exploit prior in different models.

**Improved Computational Efficiency.** We also report training time, AF prediction time, and overall runtime per BO round using T5-chem features in Fig. 21. Our method achieves the shortest training time across all datasets. While prediction time increases slightly for the largest datasets (Kinase, Laser, Photovoltaics), this overhead is offset by GP's poor performance on them. By contrast, LAPLACE incurs the highest overall cost, especially for predicting AFs during PEFT, making it the least efficient.

## 5.3. Ablation Studies

To better understand the behavior of LLMAT, evaluate the effectiveness of its individual modules, and assess its robust-

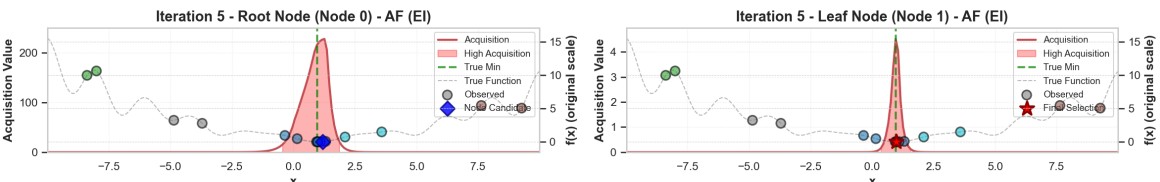

*Figure 3.* Leaf-node AFs offer more fine-grained candidate suggestions for Levy-1D optimization.

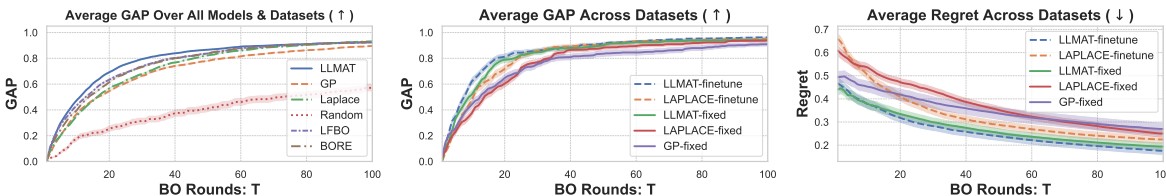

*Figure 4.* **Left:** Comparison over baselines on fixed features across all models and datasets. **Middle& Right:** Average GAP and regret (definition in 5.1) across datasets for fine-tuned (PEFT on $w$) and fixed ($w^*$) T5-chem model.

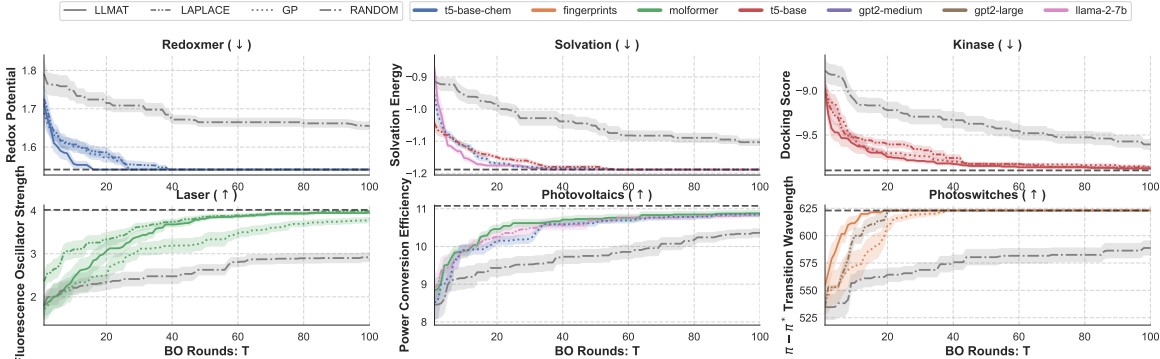

*Figure 5.* BO with fixed features (best-performing feature per algorithm shown). LLMAT achieves best overall performance, attaining top results on Solvation, Kinase, and Photoswitches using features from non-domain-specific models. The black dashed line marks the true optimum for each dataset.

ness to noise, we conduct comprehensive ablation studies. **In addition to average performance over all datasets, per-dataset results are provided in the Appendix.**

**Tree Depth Ablation.** The middle and right plots in Fig. 7 present an ablation on tree depths for LLMAT, which further strengthen the impact of refined AFs. It shows that using only LFBO ($L = 0$) performs significantly worse compared to deeper tree depths. With PEFT, performance peaks at depth 2, demonstrating the benefit of deeper partitions, while with fixed features, performance is highest at depth 1 and declines beyond due to over-partitioning.

**Cluster Selection Ablation.** We compare LLM-based clustering with K-means on feature representations across different p-values, which determine the removal of clusters with significantly poorer property values. Fig. 8 shows that LLM-based clustering achieves better average GAP and regret at p-value = 0, highlighting its superior property-awareness for BO initialization. Increasing p-values filters more clusters during AF estimation, reducing prediction time by 10–25%. While LLM-based clustering removes fewer clusters than K-means at the same p-values, its performance remains stable at 0.01 and degrades only modestly at 0.05. In contrast,

K-means is more prone to removing informative clusters, causing larger performance drops. Degradation does not always follow the p-value, as it depends on the property-cluster correlation. Fig. 32 shows LLM clustering removing irrelevant clusters on Redox-mer dataset, reducing computation and improving BO performance at the same time.

**Ablation Study for Different Algorithmic Modules.** To provide a clearer understanding of the contribution of each algorithmic component in LLMAT, we conduct a detailed ablation study by selectively removing or adding modules and evaluating their impact on overall performance, as shown in the left plot of Fig. 9. The results indicate that both MCTS and meta-learning substantially enhance performance. LLM-based clustering offers a slight improvement, while K-means has a negligible effect. This is expected, as the clustering methods are primarily designed to reduce computational cost rather than boost BO performance.

**Sensitivity of Clustering prompts.** To evaluate LLMAT's stability with respect to clustering prompts, we generated four similar prompts and compared their cluster labels agreements and cluster distributions in Fig. 10 in the Appendix. The right subplots of Fig. 9 show that the BO performance,

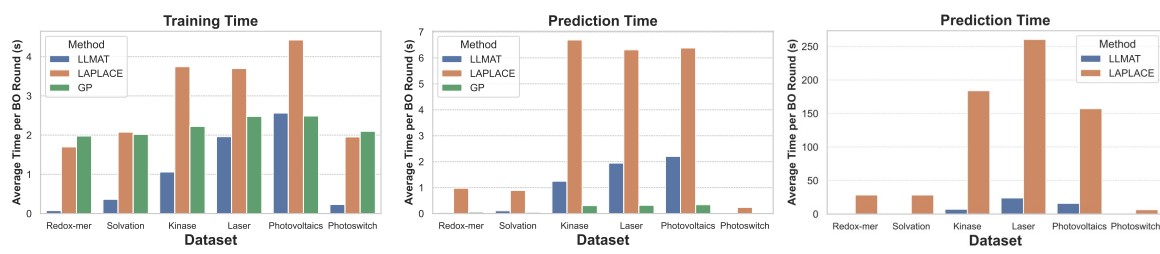

*Figure 6.* **Left & Middle:** Training and Prediction time of different methods on fixed ($w^*$) T5-Chem features. **Right:** Prediction time of LLMAT and LAPLACE for finetuning T5-chem (PEFT on $w$).

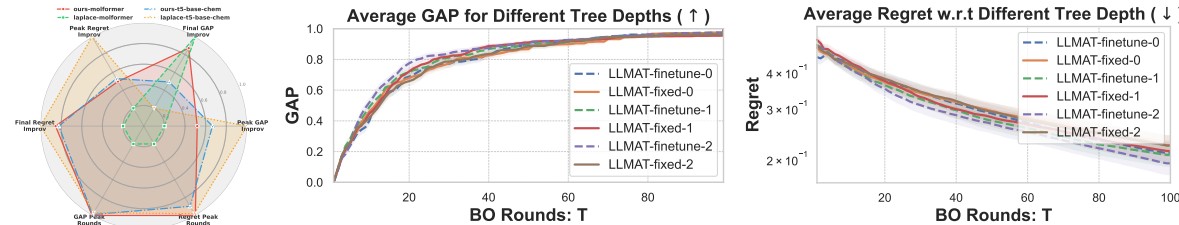

*Figure 7.* **Left:** Effectiveness of PEFT on T5-Chem and Molformer. **Middle&Right:** Tree-depth ablation on fixed and Finetuned Molformer across datasets, showing the importance of tree partition.

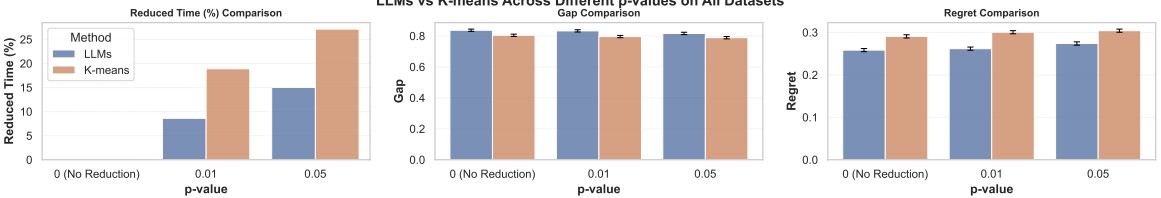

*Figure 8.* Average reduced prediction time percentage, average GAP, and average regret across six datasets for LLM-base clustering and K-means clustering w.r.t different p-values.

measured by the GAP metric, remains largely consistent. This stability can be attributed to the statistical test used during cluster selection, which incorporates observed data to enhance robustness.

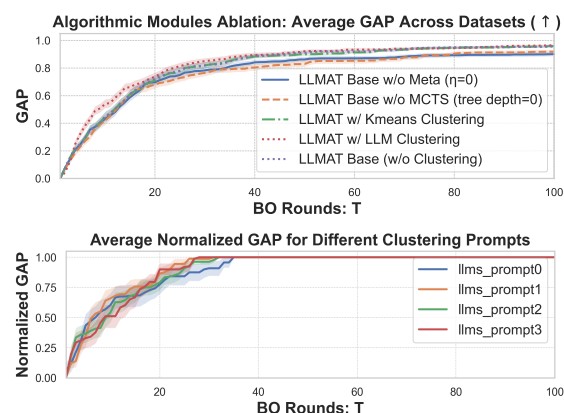

*Figure 9.* **Top:** Average GAP across six datasets illustrating the impact of individual algorithmic module ablations. **Bottom:** LLM Clustering Prompt Sensitivity in LLMAT on Redox-mer Data.

Due to space limitations, additional ablation studies on label noise, the quantile $\gamma$ and the extension to multi-objective optimization are provided in Fig. 38, Fig. 39, and Fig. 42 in the Appendix, respectively.

## 6. Conclusion

In this paper, we introduced LLMAT, a novel surrogate-free BO algorithm that incorporates LLM and foundation model features to inform the design and effectiveness of acquisition functions in BO. Our method also introduces a learned hierarchical partitioning scheme, using Monte Carlo Tree Search with tree nodes defined by shared binary classifiers to learn local acquisition functions that are informed by LLM priors. By incorporating shared parameters and meta-learning across these binary classifiers, LLMAT achieves better AF generalization in low-data regimes and enables efficient exploration of the vast search spaces of molecules. Extensive evaluations on six chemistry datasets demonstrate that LLMAT consistently outperforms baselines, highlighting the value of incorporating general LLMs and domain-specific foundation models with principled algorithmic design for molecular discovery.

**Limitations.** While evolutionary and generative algorithms exist for molecular discovery, we focus on BO ( for post-hoc verification) rather than candidate generation; a full comparison would shift focus to build benchmarks and is left for future work. Similarly, the LLM-based clustering method is prototypical and uses simple GPT-4o prompts, with extensions to other LLMs or chain-of-thought reasoning deferred.

## Impact Statement

**Potential Positive Impacts.** The proposed methods can accelerate molecular discovery by leveraging knowledge from both general LLMs and domain-specific foundation models. They can be combined with generative approaches to propose novel, unseen molecules for drug and material design, offering significant societal benefits: new drugs may save lives, and new materials could contribute to environmental protection.

**Potential Negative Impacts.** However, the method also carries potential risks. For example, the same capabilities could be misused to create toxic or harmful compounds, and overreliance on AI predictions without rigorous experimental validation could lead to unintended consequences. Responsible use, careful oversight, and appropriate safety measures are therefore critical.

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

## A. Additional Background

### A.1. Indirect Estimation of Acquisition Functions

Based on the definition of the acquisition function in Eq.(1), it's natural to first calculate the predictive posterior $p(y|x, \mathbf{D}_t)$, then estimate the AFs given the selected utility functions. In the following, we introduce two typical surrogate models that are non-parametric and parametric, respectively.

**Gaussian Process (GP).** A typical choice of a non-parametric surrogate model $\hat{f}(x)$ is a GP with $p(\hat{f}(x)|\mathbf{D}_t) = \mathcal{N}(\mathbf{k}_t^T \mathbf{C}_t^{-1} \mathbf{y}_t, k(x, x) - \mathbf{k}_t^T \mathbf{C}_t^{-1} \mathbf{k}_t)$ being Gaussian given some kernel function $k(\cdot, \cdot)$, where $\mathbf{k}_t[i] = k(x_i, x) \forall i \in [t]$, $\mathbf{K}_t[i, j] = k(x_i, x_j), \forall i, j \in [t]$, and $\mathbf{C}_t = \mathbf{K}_t + \sigma^2 \mathbf{I}_t$. In this case, the predictive posterior is tractable and has the form of $p(y|x, \mathbf{D}_t) = \mathcal{N}(\mathbf{k}_t^T \mathbf{C}_t^{-1} \mathbf{y}_t, k(x, x) + \sigma^2 - \mathbf{k}_t^T \mathbf{C}_t^{-1} \mathbf{k}_t)$. However, the basic GPs have a $\mathcal{O}(t^3)$ computational complexity for training, even sparse GPs still require $\mathcal{O}(t)$, posing difficulty to scale to a large dataset of observations.

**Bayesian Neural Network (BNN).** If we consider parametric surrogate model $\hat{f}(x) = \text{NN}(\theta, x)$ with $\text{NN}(\theta, \cdot)$ being a pre-defined Neural Network (NN) of parameter $\theta$, then, the predictive posterior of BNN is $p(y|x, \mathbf{D}_t) = \int p(y|x, \theta)p(\theta|\mathbf{D}_t)d\theta$. The posterior $p(\theta|\mathbf{D}_t)$ is often not analytically tractable. Hence, solutions like Laplace Approximation (LA), Variational Inference (VI) and Monte Carlo Sampling are often adopted to approximate the integral.

### A.2. Parameter-Efficient Finetuning

Fine-tuning large pretrained models on downstream tasks often requires updating all model parameters, which can be computationally expensive and memory-intensive. Parameter-efficient fine-tuning (PEFT) methods aim to reduce this overhead by introducing a small set of trainable parameters while keeping the original pretrained model weights frozen.

Among various PEFT methods, *Low-Rank Adaptation* (LoRA) (Hu et al., 2021) has gained popularity due to its efficiency and effectiveness. LoRA hypothesizes that the update to the pretrained weight matrices can be approximated by a low-rank decomposition. Formally, consider a pretrained weight matrix $W_0 \in \mathbb{R}^{d \times k}$ in a neural network (e.g., a linear layer or attention projection). Instead of updating $W_0$ directly, LoRA introduces two trainable matrices $A \in \mathbb{R}^{d \times r}$ and $B \in \mathbb{R}^{r \times k}$, where $r \ll \min(d, k)$ is the rank of the adaptation:

$$W = W_0 + \Delta W = W_0 + BA,$$

where $BA$ is a low-rank matrix capturing task-specific updates. During fine-tuning, $W_0$ is kept frozen, and only $A$ and $B$ are optimized. For an input $x \in \mathbb{R}^k$, the layer output becomes

$$y = Wx = W_0 x + BAx.$$

To further control the magnitude of updates, a scaling factor $\alpha$ is often introduced:

$$W = W_0 + \frac{\alpha}{r} BA,$$

where the factor $\frac{\alpha}{r}$ ensures the update has a similar scale across different rank settings. This formulation significantly reduces the number of trainable parameters from $d \times k$ to $r(d + k)$ while retaining strong adaptation capability.

### A.3. Welch's ANOVA

Suppose we have $k$ clusters, where cluster $i$ has sample size $n_i$, sample mean $\bar{x}_i$, and sample variance $\sigma_i^2$. Define the weights:

$$w_i = \frac{n_i}{\sigma_i^2}, \quad i = 1, \ldots, k,$$

and the weighted mean:

$$\bar{x}_w = \frac{\sum_{i=1}^{k} w_i \bar{x}_i}{\sum_{i=1}^{k} w_i}.$$

The Welch F-statistic is then

$$F = \frac{\sum_{i=1}^{k} w_i (\bar{x}_i - \bar{x}_w)^2}{(k-1) \left[ 1 + \frac{2(k-2)}{k^2-1} \sum_{i=1}^{k} \frac{(1 - w_i / \sum_j w_j)^2}{n_i - 1} \right]}.$$

The approximate degrees of freedom for the denominator is given by the Satterthwaite approximation:

$$\nu \approx \frac{k^2 - 1}{3 \sum_{i=1}^{k} \frac{(1 - w_i / \sum_j w_j)^2}{n_i - 1}}.$$

Finally, $F$ is compared against an $F$-distribution with $(k-1, \nu)$ degrees of freedom to compute the p-value.

### A.4. Games-Howell Post-hoc Test

Suppose we have $k$ clusters, where cluster $i$ has sample size $n_i$, sample mean $\bar{x}_i$, and sample variance $\sigma_i^2$. For a pair of groups $(i, j)$, the test statistic is

$$t_{ij} = \frac{\bar{x}_i - \bar{x}_j}{\sqrt{\frac{\sigma_i^2}{n_i} + \frac{\sigma_j^2}{n_j}}}.$$

The degrees of freedom for this comparison are approximated using the Welch–Satterthwaite equation:

$$\nu_{ij} = \frac{\left( \frac{\sigma_i^2}{n_i} + \frac{\sigma_j^2}{n_j} \right)^2}{\frac{(\sigma_i^2 / n_i)^2}{n_i - 1} + \frac{(\sigma_j^2 / n_j)^2}{n_j - 1}}.$$

The p-value is computed from the $t$-distribution with $\nu_{ij}$ degrees of freedom:

$$p_{ij} = 2 \cdot \left( 1 - T_{\nu_{ij}}(|t_{ij}|) \right),$$

where $T_{\nu_{ij}}$ is the cumulative distribution function of the $t$-distribution. Optionally, a multiple comparison correction (e.g., Bonferroni) can be applied.

## B. Related Works

### B.1. LLMs for Optimization.

Recent research has increasingly leveraged pretrained Large Language Models (LLMs) as informative priors in optimization tasks across a variety of domains. For instance, LLMs have been used to improve prompt optimization strategies by conditioning on query-dependent representations (Sun et al., 2024; Yang et al., 2024; Guo et al., 2024). Other works have integrated LLMs into evolutionary algorithms, using them as generative operators to evolve prompts or candidate solutions (Meyerson et al., 2024; Lehman et al., 2022; Chen et al., 2023a).In the context of Bayesian Optimization (BO), LLMs have also been employed to guide acquisition functions, either for hyperparameter tuning (Liu et al., 2024b; Zhang et al., 2023) or for scientific discovery tasks such as molecular optimization (Kristiadi et al., 2024; Ramos et al., 2023).

### B.2. High-dimensional BO

Several approaches have been developed to scale BO to high-dimensional search spaces, including structural assumptions, subspace embedding, variable selection, and local modeling and space partitioning. Wang et al. (2020; 2021) progressively bifurcate the search space into more promising and less promising subspaces, starting with the full candidate space $\Omega$. K-means clustering is applied to observed data (features $x$ and values $y$) within each space to split the data into "good" and "bad" clusters based on the average values of the clusters. These labeled clusters are then used to train a Support Vector Machine (SVM) classifier, which produces a non-linear decision boundary to define latent actions for bifurcating the search space. Finally, local surrogate models for BO are trained on observed data in the selected subspace via Upper Confidence Bound (UCB). Based on Wang et al. (2020), Li et al. (2025) propose to reweigh the samples in different partition by their UCB values without constraining sampling to certain partitions. In contrast, we do not separately use SVMs for partitioning and additional models for acquisition function estimation.

We focus on leveraging LLM-derived priors to enhance high-dimensional Bayesian Optimization (BO) for molecular discovery by using LLMs for feature extraction, finetuning and prompting for clustering. Unlike prior work, we introduce a shared neural network classifier that simultaneously partitions the candidate space and estimates acquisition functions at each tree node, effectively amortizing both tasks. To further improve data efficiency, we adopt a meta-learning (Chen et al., 2021; 2023b) approach that amortizes classifier training across nodes, enabling more stable learning in early data-scarce stages. Moreover, while previous methods assume a continuous search space and rely on rejection sampling, they fail in discrete settings where selected regions may contain no valid candidates. We address this limitation via a backtracking strategy. Our method traverses a UCB-guided path through the search tree, estimating local acquisition functions from leaf to parent until valid candidates are found. In the worst case, it returns to the original Likelihood-Free BO (LFBO) approach.

## C. Partition Selection Algorithm

---

**Algorithm 2** Partition Selection for Refined and Reduced AFs

---

1: **Input:** Dataset $\mathbf{D}_t$, tree depth $L$, candidates in selected clusters $\Omega_{\text{cluster}}$, classifier $\theta$, learning rate $\eta$, SGD steps $K$, quantile $\gamma$, UCB hyper-parameter $\lambda$, time budget $T$, all nodes $\mathcal{N}$.
2: **for** $t = 0$ **to** $T$ **do**
3:    Run Algo. 1;
4:    #Select the search tree path.
5:    Set $k = 0$, $\tau_k = \Phi_k^{-1}(\gamma)$;
6:    Path $\mathcal{P} \leftarrow []$;
7:    **while** $k \in \mathcal{N}$ **do**
8:       $\mathcal{P} \leftarrow \mathcal{P} + [k]$;
9:       Compute partition score for $2k + 1$ and $2k + 2$;
10:      $k = \arg\max_{j \in \{2k+1, 2k+2\}} \mathcal{S}_j$;
11:    **end while**
12:    #Backtrack and select local AF up the path.
13:    $\mathcal{P} \leftarrow \text{Reverse}(\mathcal{P})$;
14:    selected = 0;
15:    **for** each $k$ in $\mathcal{P}$ **do**
16:       **if** $|\Omega_k \cap \Omega_{\text{cluster}}| = 0$ **then**
17:          continue;
18:       **else**
19:          selected = k;
20:          break;
21:       **end if**
22:    **end for**
23:    Sample $x_{t+1} = \arg\max_{x \in \Omega_{\text{selected}} \cap \Omega_{\text{cluster}}} \alpha(x; \Omega_{\text{selected}} \cap \mathbf{D}_t, \tau_{\text{selected}})$;
24:    Evaluate $y_{t+1} = f(x_{t+1})$;
25:    Update $\mathbf{D}_{t+1} \leftarrow \mathbf{D}_t \cup \{(x_{t+1}, y_{t+1})\}$;
26: **end for**

---

## D. Claims

**Precise Use of Large Language Models (LLMs)** We use LLMs as grammar correction tools and for improving written text flow, combined with human proofreading. No LLM was used for the idea or primary text of the paper. However, as this paper investigates how LLMs can help inform acquisition functions for BO over molecular datasets, we incorporate LLMs as algorithmic modules, where we also prompt chatGPT to generate proper LLM-clustering prompts.

# E. Clustering Prompts

## E.1. Redoxmer

---

**Redoxmer**

You are a chemist, please use the molecules provided, group them into five clusters based on their redox potential.

> **Clustering Scale: extremely low: 0, low: 1, medium: 2, high: 3, extremely high: 4**

**Analyze the following features for each molecule:**

- Number and type of electron-withdrawing groups (e.g., $CF_3$, $NO_2$, CN, halogens)

- Number and type of electron-donating groups (e.g., alkyl, methoxy, hydroxyl)

- Positioning of substituents on aromatic rings (meta, para, ortho)

- Presence of sulfur-containing functional groups (e.g., $SCF_3$, S=O)

- Degree of molecular polarity (based on F, O, or N atoms)

Use these criteria to evaluate the redox potential qualitatively and cluster the molecules accordingly.

> **Response Format:** *Respond strictly with the counter and numerical cluster labels only. Do not include any additional text.*

**Now please cluster the following molecules:**

> {molecules to be inserted here}

---

## E.2. Solvation

---

**Solvation**

You are a chemist, please use the molecules provided, group them into five clusters based on predicted solvation energy.

> **Clustering Scale: extremely low: 0, low: 1, medium: 2, high: 3, extremely high: 4**

**Consider these molecular features:**

- Polarity and hydrogen bonding capability

- Molecular size and surface area

- Number and type of charged/ionizable groups

- Hydrophilic/hydrophobic balance

Use these criteria to evaluate the solvation energy qualitatively and cluster the molecules accordingly.

> **Response Format:** *Respond strictly with the counter and numerical cluster labels only. Do not include any additional text.*

**Now please cluster the following molecules:**

> {molecules to be inserted here}

---

### E.3. Kinase

> ## Kinase
>
> Act as a computational chemist. You have a list of SMILES strings for molecules, and I want to group them into 5 clusters (0 to 4) where cluster 0 has the lowest predicted kinase docking affinity and cluster 4 has the highest.
>
> > **Clustering Scale: extremely low: 0, low: 1, medium: 2, high: 3, extremely high: 4**
>
> **Prioritize these criteria:**
>
> - Presence of kinase-binding motifs (e.g., hinge-binding heterocycles, hydrophobic pockets)
>
> - Functional groups (e.g., hydrogen bond donors/acceptors, aromatic rings)
>
> - Molecular weight and polarity (smaller/lipophilic molecules often bind kinases better)
>
> - Similarity to known kinase inhibitors (e.g., ATP analogs, tyrosine kinase inhibitors)
>
> > **Response Format:** *Respond strictly with the counter and numerical cluster labels only. Do not include any additional text.*
>
> **Now please cluster the following molecules:**
>
> > {molecules to be inserted here}

### E.4. Photovoltaics

> ## Photovoltaics
>
> You are a chemist, please use the molecules provided, group them into five clusters based on predicted photovoltaic conversion efficiency.
>
> > **Clustering Scale: extremely low: 0, low: 1, medium: 2, high: 3, extremely high: 4**
>
> **Consider these molecular features:**
>
> - Electronic structure indicators (conjugation extent, aromatic systems, electron-rich/deficient regions, push-pull molecular design)
>
> - Molecular architecture (planarity potential, $\pi$-system connectivity, structural rigidity, molecular size)
>
> - Light harvesting features (conjugated backbone length, donor-acceptor patterns, chromophore presence, substituent effects)
>
> Use these criteria to evaluate the photovoltaic conversion efficiency qualitatively and cluster the molecules accordingly.
>
> > **Response Format:** *Respond strictly with the counter and numerical cluster labels only. Do not include any additional text.*
>
> **Now please cluster the following molecules:**
>
> > {molecules to be inserted here}

## E.5. Laser

> ## Laser
>
> Act as a computational chemist with expertise in photophysics. Group these SMILES strings into 5 clusters based on predicted fluorescence oscillator strength (relevant for lasers).
>
> > **Clustering Scale: very low: 0, low: 1, moderate: 2, high: 3, very high: 4**
>
> **Consider these factors:**
>
> - Conjugation length: Longer conjugation increases oscillator strength
>
> - Aromaticity: Aromatic systems often have strong $\pi$-$\pi^*$ transitions
>
> - Functional groups: Electron-donating/withdrawing groups alter oscillator strength
>
> - Molecular rigidity: Rigid molecules tend to have higher oscillator strength
>
> > *Response Format: Respond strictly with the counter and numerical cluster labels only. Do not include any additional text.*
>
> **Now please cluster the following molecules:**
>
> > {molecules to be inserted here}

## E.6. Photoswitches

> ## Photoswitches
>
> Act as a computational chemist with expertise in photochemistry. You have a list of SMILES strings for organic molecules, and want to group them into 5 clusters based on their predicted $\pi$-$\pi^*$ transition wavelengths.
>
> > **Clustering Scale:**
> >
> > 1. **Cluster 0: deep UV range (200–300 nm)**
> >
> > 2. **Cluster 1: UV range (300–400 nm)**
> >
> > 3. **Cluster 2: blue/visible range (400–500 nm)**
> >
> > 4. **Cluster 3: green/red/visible range (500–700 nm)**
> >
> > 5. **Cluster 4: near-infrared range (700–1000 nm)**
>
> Use your knowledge of molecular structure and photochemistry to analyze the SMILES strings and assign cluster labels. **Consider the following factors:**
>
> - **Conjugation length:** Longer conjugation typically shifts the $\pi$-$\pi^*$ transition to longer wavelengths
>
> - **Aromaticity:** Aromatic systems often have $\pi$-$\pi^*$ transitions in the UV/visible range
>
> - **Functional groups:** Electron-donating or electron-withdrawing groups can alter the HOMO-LUMO gap
>
> - **Molecular planarity:** Planar molecules tend to have stronger $\pi$-$\pi^*$ transitions
>
> > *Response Format: Respond strictly with the counter and numerical cluster labels only. Do not include any additional text.*
>
> **Now please cluster the following molecules:**
>
> > {molecules to be inserted here}

### E.7. Variations in Prompt

We ask ChatGPT to generate 4 alternative prompts based on our original prompt, the cluster label disagreements, the label distributions, and the generated prompts are listed below.

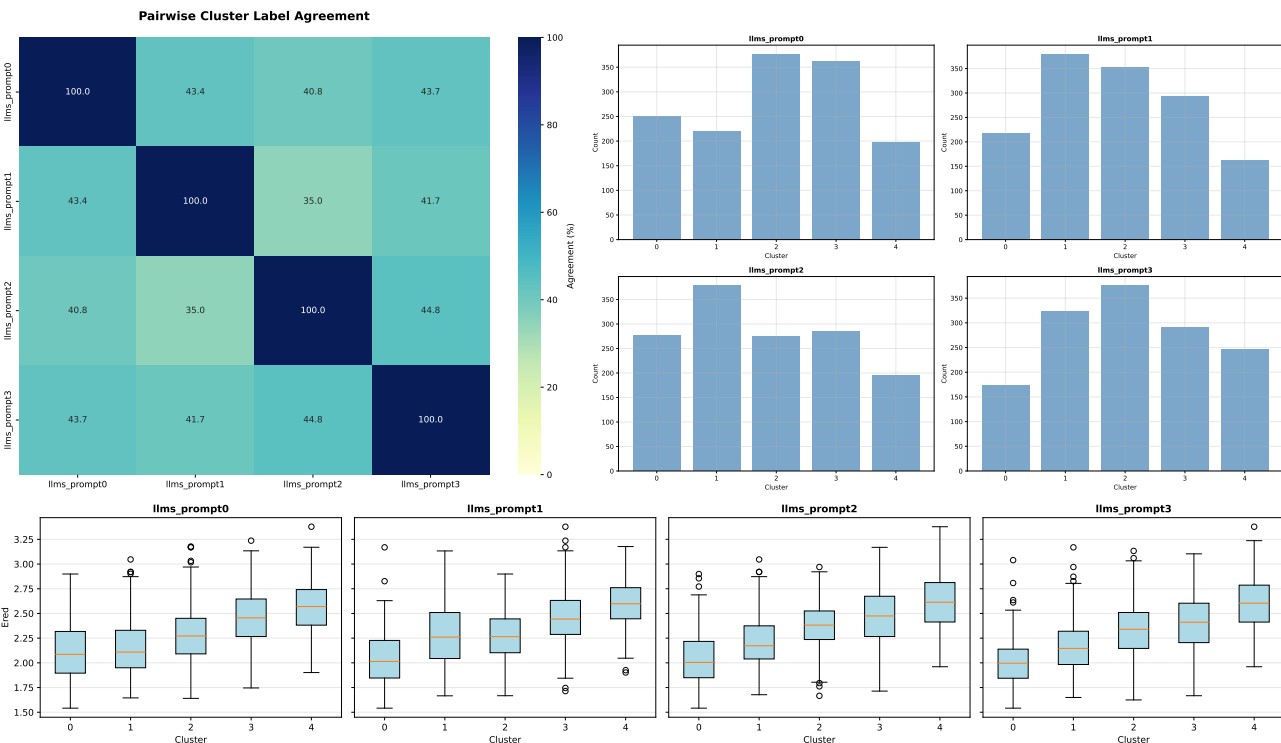

*Figure 10.* Prompt variation's effect on cluster labels.

---

**Prompt0**

You are a chemist, please use the molecules provided, group them into five clusters based on their redox potential (extremely low : 0, low: 1, medium: 2, high: 3, extremely high :4).
**Analyze the following features for each molecule:**

- Number and type of electron-withdrawing groups (e.g., $CF_3$, $NO_2$, CN, halogens).

- Number and type of electron-donating groups (e.g., alkyl, methoxy, hydroxyl).

- Positioning of substituents on aromatic rings (meta, para, ortho).

- Presence of sulfur-containing functional groups (e.g., $SCF_3$, S=O).

- Degree of molecular polarity (based on F, O, or N atoms).

Use these criteria to evaluate the redox potential qualitatively and cluster the molecules accordingly.

> **Response Format:** *Respond strictly with the counter and numerical cluster labels only. Do not include any additional text.*

**Now please cluster the following molecules:**

> {molecules to be inserted here}

## Prompt1

Act as a chemist. Using the given molecules, assign each one to a redox-potential cluster: 0 = extremely low, 1 = low, 2 = medium, 3 = high, 4 = extremely high. **Evaluate each molecule based on:**

- Electron-withdrawing substituents ($CF_3$, $NO_2$, CN, halogens, etc.)

- Electron-donating substituents (alkyl, OMe, OH, etc.)

- Substituent positions on aromatic systems (ortho/meta/para)

- Sulfur-containing functional groups ($SCF_3$, sulfoxides, etc.)

- Overall polarity from heteroatoms (F/O/N)

Use these qualitative features to estimate redox strength and cluster the molecules.

> **Response Format:** *Respond strictly with the counter and numerical cluster labels only. Do not include any additional text.*

**Now please cluster the following molecules:**

> {molecules to be inserted here}

## Prompt2

You are an expert computational chemist. For each provided molecule, perform the following steps: Identify electron-withdrawing and electron-donating groups. Determine substituent positions on aromatic cores. Note any sulfur-based functionalities (e.g., $SCF_3$, S=O). Assess molecular polarity based on heteroatom composition. Infer the molecule's qualitative redox potential from these structural features. Then group all molecules into five redox-potential clusters (0–4) from extremely low to extremely high.

> **Response Format:** *Respond strictly with the counter and numerical cluster labels only. Do not include any additional text.*

**Now please cluster the following molecules:**

> {molecules to be inserted here}

## Prompt3

As a chemist specializing in structure–property relationships, analyze each molecule and justify its qualitative redox potential. Consider:

- Strength and number of EWG vs EDG

- Aromatic substitution pattern and resonance effects

- Sulfur-functional group contributions

- Polarity arising from heteroatoms

Based on the identified features, explain your reasoning and categorize each molecule into one of five redox classes (0: extremely low → 4: extremely high).

> **Response Format:** *Respond strictly with the counter and numerical cluster labels only. Do not include any additional text.*

**Now please cluster the following molecules:**

> {molecules to be inserted here}

# F. Experimental Details

We follow most of the settings in Kristiadi et al. (2024) for using the foundation models and baselines: GP and Laplace, while we Refactored their code presented in repo `https://github.com/wiseodd/lapeft-bayesopt` for better extension.

**Computational Resource**   The experiments were conducted on multiple server nodes, each equipped with 6 CPUs and a single GPU with 32 GB of memory. To obtain results faster, the experiments were run across different clusters. We ensured that, for each dataset, all algorithms were tested on the same type of CPU and GPU.

## F.1. Foundation Models

**Features and Prompts for Foundation Model.**   For the LLM features, we average the last transformer embeddings along the sequence dimension, ignoring padding and EOS tokens. All LLM-related components in this work were implemented using the Hugging Face Transformers library (Wolf et al., 2019). We use the single SMILES string as the prompt input for feature extraction and parameter-efficient finetuning of these foundation models.

## F.2. BO on Fixed-features

We use the same batch size 256 for AFs estimation (*i.e.*, prediction) for all the algorithms.

### F.2.1. TRAINING DETAILS

**GP**   For GP, we use BoTorch (Balandat et al., 2020), with the Tanimoto kernel from Gauche (Griffiths et al., 2023). The marginal likelihood is optimized using Adam (Kingma & Ba, 2014) with a learning rate of 0.01 for 500 epochs.

**Laplace**   A 2-hidden-layer multilayer perceptron with 50 units per layer is used. The network is optimized with Adam at a learning rate of $1 \times 10^{-3}$ and weight decay $5 \times 10^{-4}$ for 500 epochs with batch size 20, using cosine annealing for the learning rate (Loshchilov & Hutter, 2016). The Laplace approximation is applied post hoc, with prior precision tuned via marginal likelihood for 100 iterations. The Hessian is approximated using a Kronecker structure (Ritter et al., 2018).

**LLMAT**   We build the MCTs that satisfy the tree depth and minimal leaf sample size constraints. Then we train classifiers that are 2-hidden-layer multilayer perceptrons with 50 units per layer using ReLU activation. The classifiers are trained with Adam of learning rate $1 \times 10^{-2}$ and weight decay $5 \times 10^{-4}$ for 50 epochs with batch size 256. Other hyperparameters like the quantile $\gamma$, the $\lambda$ for UCB, the meta-learning rate $\eta$, the threshold for partitioning, the tree depth, and the p-value are summarized in Tab. 1.

*Table 1.* Hyperparameter Configuration

| Category | Parameter | Value |
|---|---|---|
| **General** | $\gamma$ | 0.5 |
| | $\lambda$ | 0.5 |
| | $\eta$ | 0.005, 0.01 |
| | tree_depth | 3 |
| | p_val | 0, 0.01, 0.05 |
| | threshold | 0.5 |
| **Fix Args** | batch_size | 256 |
| | *Head Parameters:* | |
| | n_epochs | 50 |
| | learning rate | 1e-2 |
| | batch_size | 256 |
| | leaf_sample_size | 2 |

### F.3. BO with PEFT

We keep the same LoRA configuration for our algorithm and the Laplace baseline. The batch size for acquisition function estimation was set to 16 for T5-Chem and 32 for Molformer. Same batch size was also used for LoRA training.

**LoRA Configuration.** We used LoRA with rank 4, applied without bias on the key and value attention weights. The scaling factor $\alpha$ was set to 16, and dropout with probability 0.1 was applied. We followed the implementation from HuggingFace's PEFT library (Mangrulkar et al., 2022).

#### F.3.1. TRAINING DETAILS OF LAPLACE

In Kristiadi et al. (2024), the following setting was applied to PEFT with Laplace Approximation.

**LoRA Training.** The LoRA parameters and the regression head were jointly trained using AdamW with learning rates of $3 \times 10^{-4}$ and $1 \times 10^{-3}$ for the LoRA and regression head weights, respectively (except for the Photoswitch dataset, where they used $3 \times 10^{-3}$ and $1 \times 10^{-2}$). Training was performed for 50 epochs with weight decay 0.01. Subsequently, the regression head was optimized for 100 epochs under the same hyperparameters.

**Laplace Approximation.** The Laplace approximation was applied to both the LoRA and regression head weights. We used a Kronecker-factored Hessian and optimized the layerwise prior precisions with post hoc marginal likelihood for 200 iterations, following Daxberger et al. (2021).

#### F.3.2. TRAINING DETAILS OF OUR ALGORITHM

Following the fixed-feature setting, we construct MCTs subject to tree-depth and minimum leaf-sample constraints, and train 2-layer MLP classifiers (50 hidden units per layer, ReLU activations). To enable larger classifier batch sizes, batching is applied after the feature extractor and LoRA layers. Classifiers are trained with Adam (lr $1 \times 10^{-2}$, weight decay $5 \times 10^{-4}$, batch size 256, 50 epochs). LoRA learning rates are task-specific: $3 \times 10^{-5}$ (Solvation, Kinase), $5 \times 10^{-5}$ (Redoxmer, Laser), and $1 \times 10^{-7}$ (Photovoltaics, Photoswitch). Additional hyperparameters, including $\gamma$, $\lambda$, $\eta$, partitioning threshold, tree depth, and $p$-value, are listed in Tab. 1.

*Table 2.* Hyperparameter Configuration

| Category | Parameter | Value |
|---|---|---|
| **PEFT** | batch_size | 16, 32 |
| | *Head Parameters:* | |
| | n_epochs | 50 |
| | learning rate | 1e-2 |
| | batch_size | 256 |
| | leaf_sample_size | 2 |
| | *LoRA Parameters:* | |
| | n_epochs | 50 |
| | learning rate | 3e-5, 5e-5, 1e-7 |
| | batch_size | 16, 32 |

# G. Additional Experimental Results

In this section, we present additional experimental results on historical optimums, GAP metrics, regret, and computational costs for each dataset based on T5-chem and Molformer.

## G.1. Fixed-feature results for more foundation models

### G.1.1. HISTORICAL OPTIMUMS

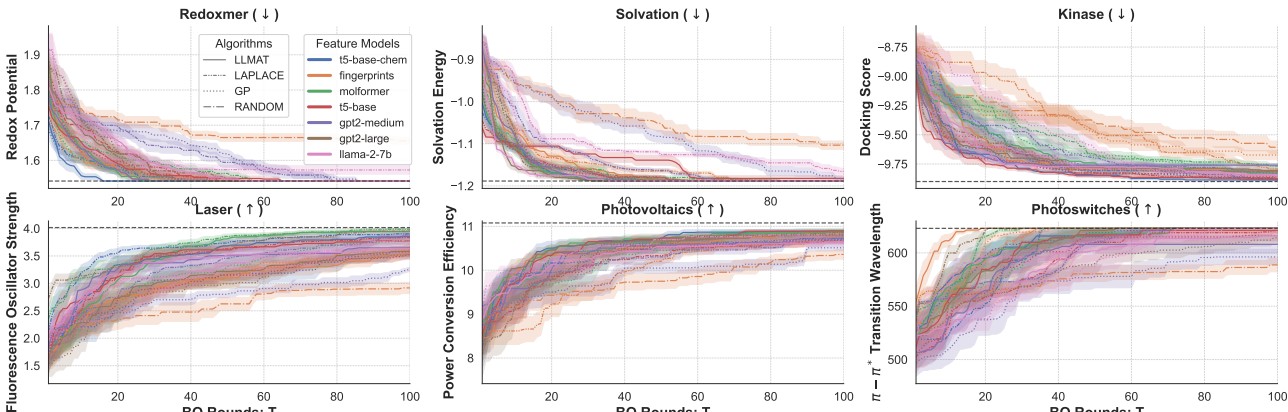

*Figure 11.* BO process for different datasets on features extracted from various foundation models.

### G.1.2. COMPUTATION TIME

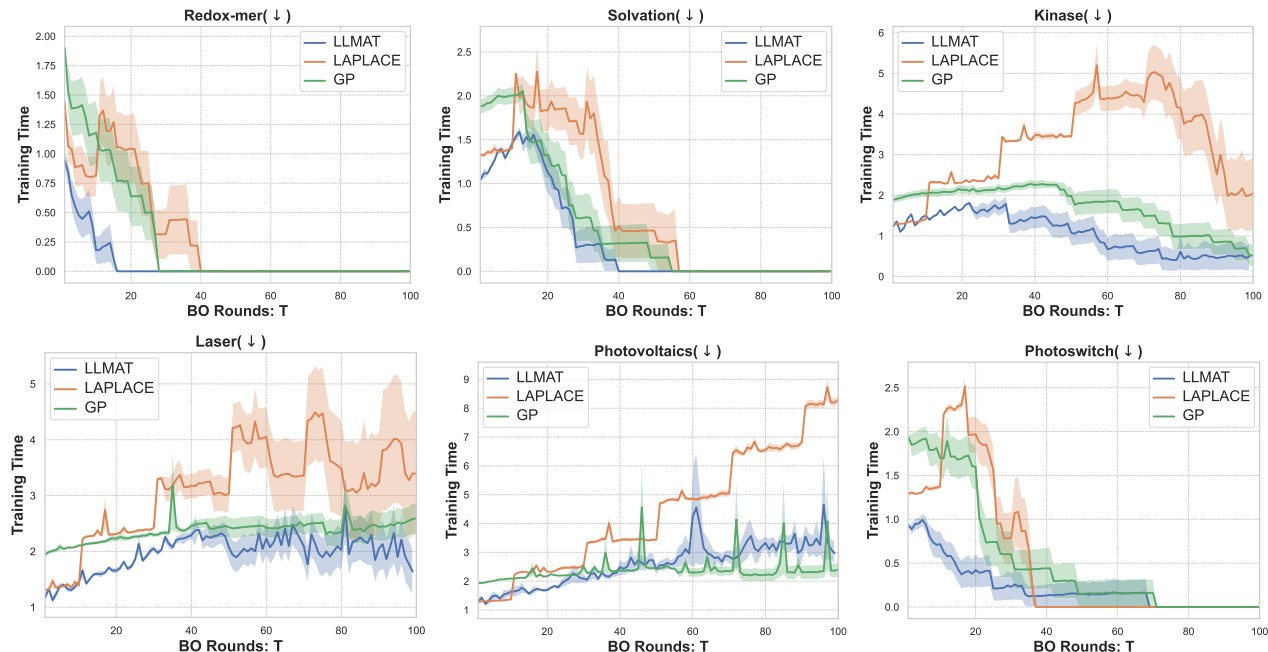

*Figure 12.* Training time comparison on different chemistry datasets with T5-Chem model.

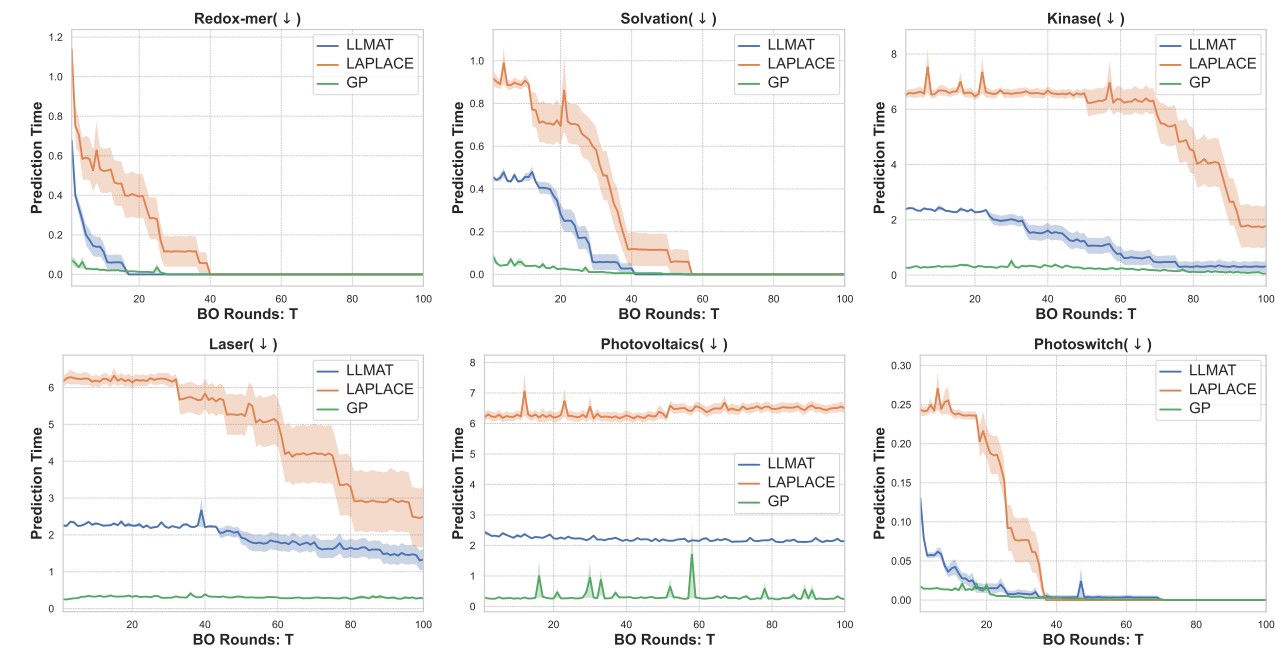

*Figure 13.* Prediction time comparison on different chemistry datasets with T5-Chem model.

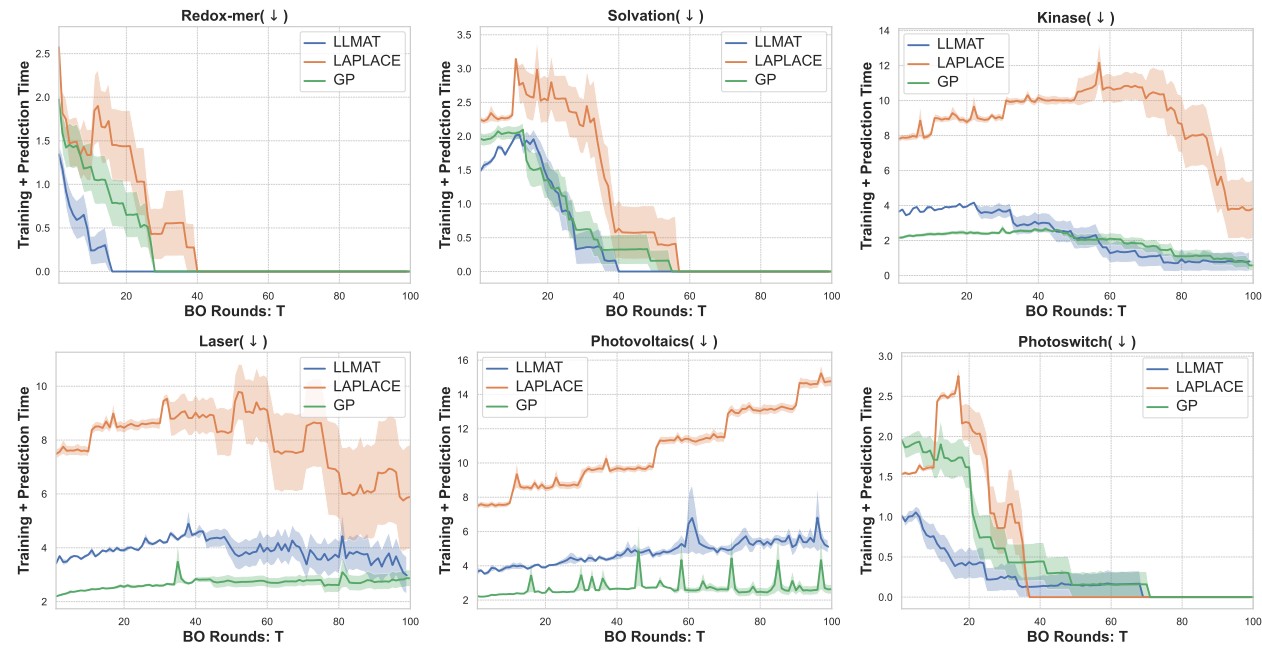

*Figure 14.* Overall time comparison on different chemistry datasets with T5-Chem model.

## G.2. Fixed and finetuned results for T5-chem Model

### G.2.1. HISTORICAL OPTIMUMS

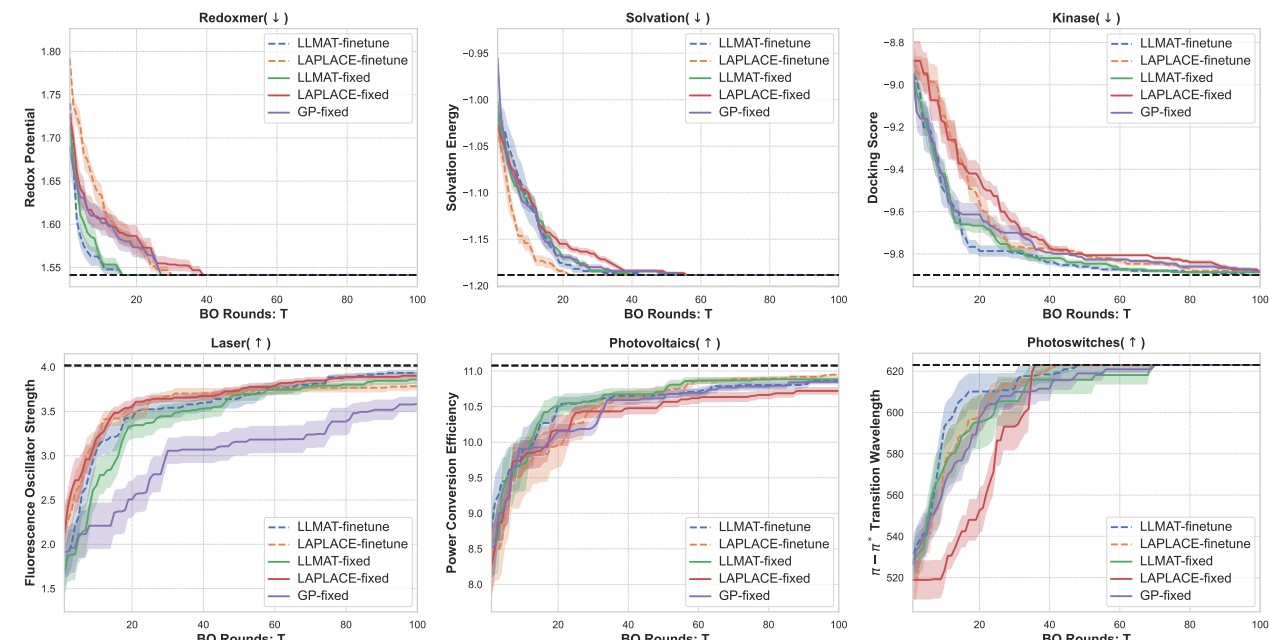

*Figure 15.* Historical optimums comparison on different chemistry datasets using T5-Chem model.

### G.2.2. GAP METRIC

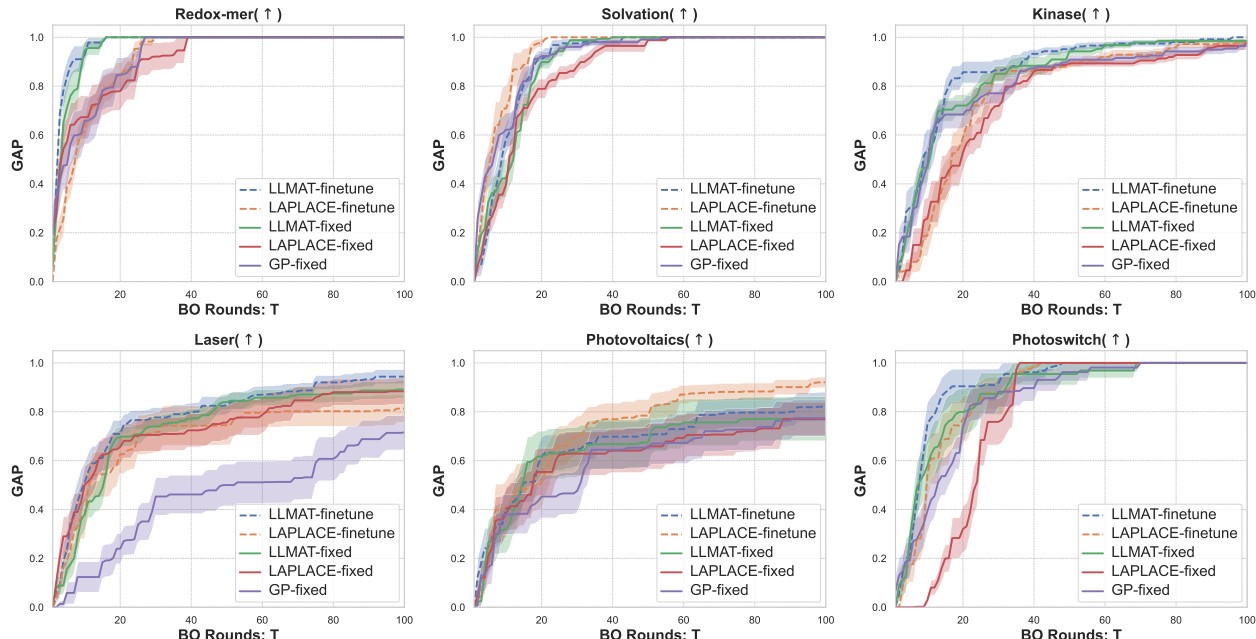

*Figure 16.* GAP comparison on different chemistry datasets with T5-Chem model.

### G.2.3. AVERAGE REGRET

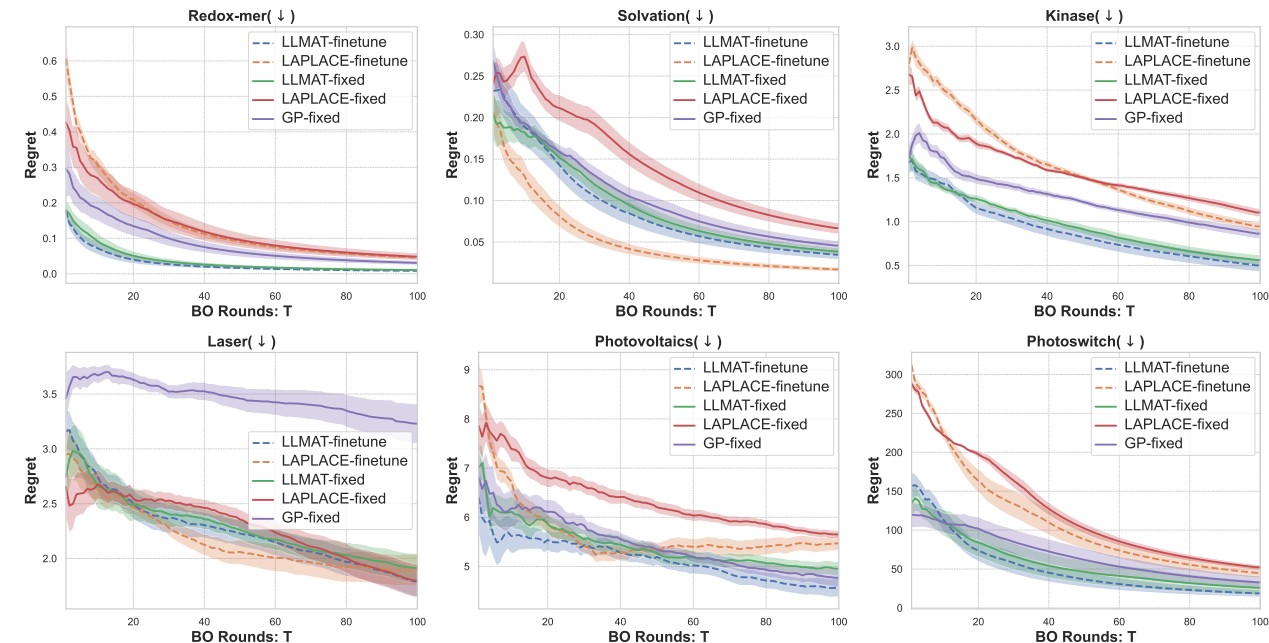

*Figure 17.* Regrets comparison on different chemistry datasets using T5-Chem model.

### G.2.4. COMPUTATION TIME FOR PEFT

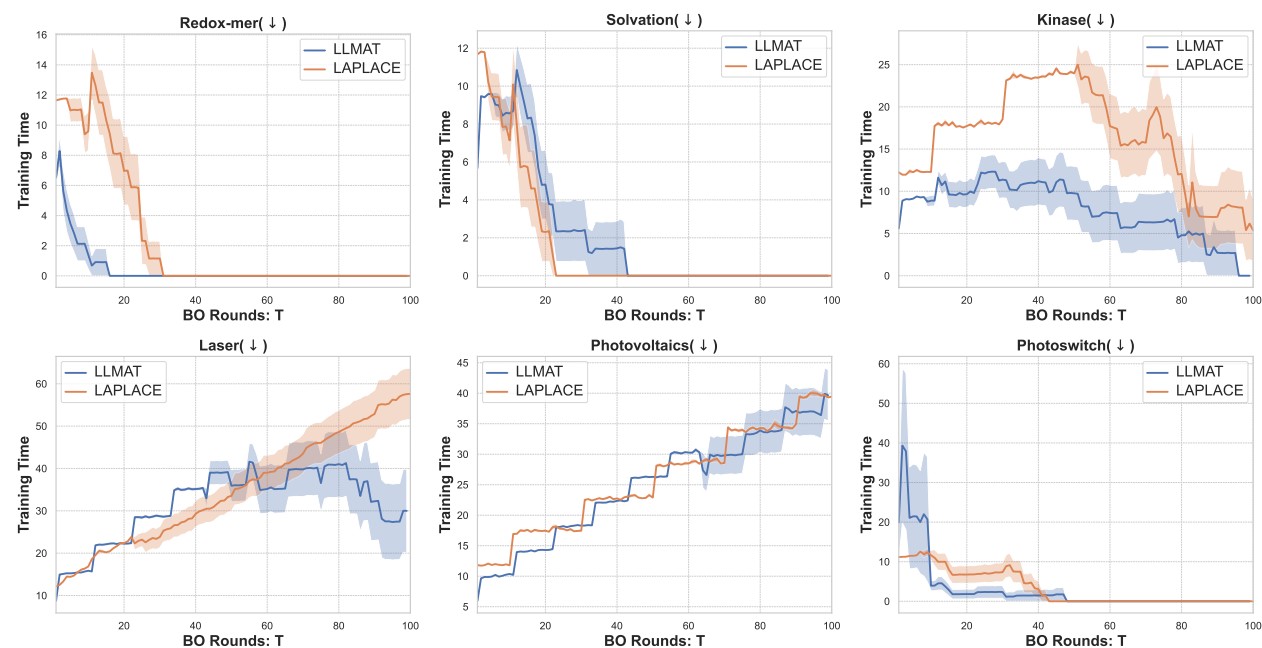

*Figure 18.* Training time comparison on different chemistry datasets with T5-Chem model.

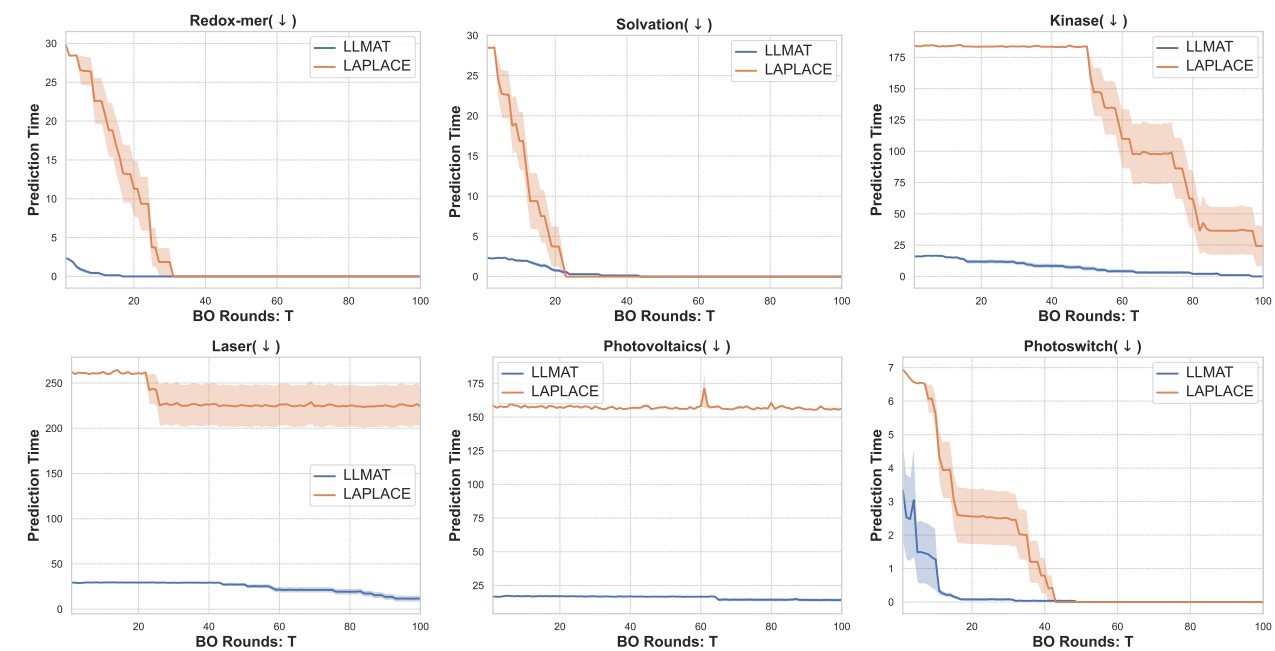

*Figure 19.* Prediction time comparison on different chemistry datasets with T5-Chem model.

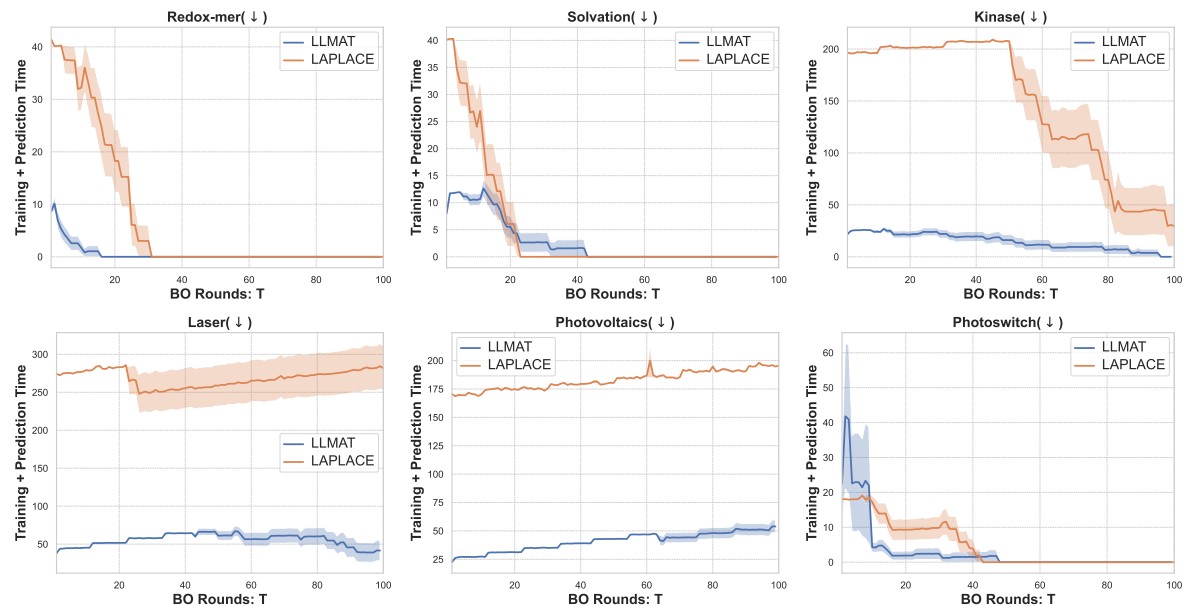

*Figure 20.* Overall time comparison on different chemistry datasets with T5-Chem model.

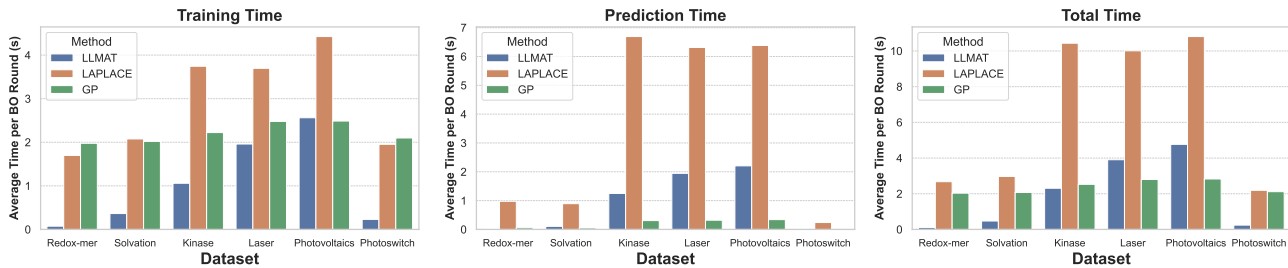

*Figure 21.* Computation time of different algorithms with fixed T5-Chem features across datasets.

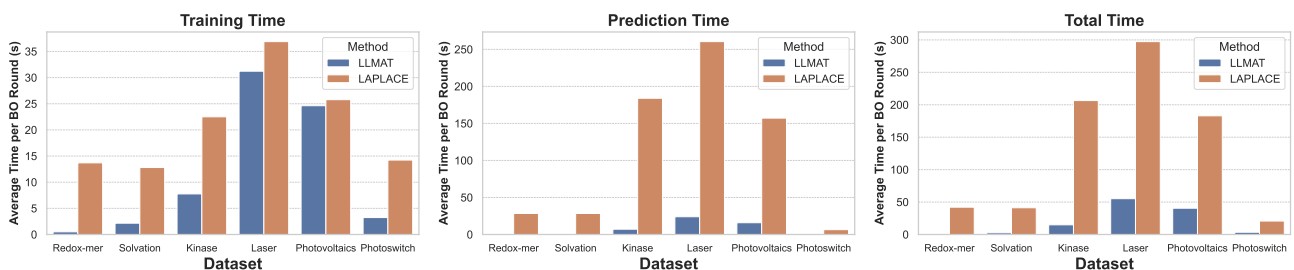

*Figure 22.* BO time for different datasets on finetuning T5-Chem model with LoRA.

## G.3. Fixed and finetuned results for Molformer

### G.3.1. HISTORICAL OPTIMUMS

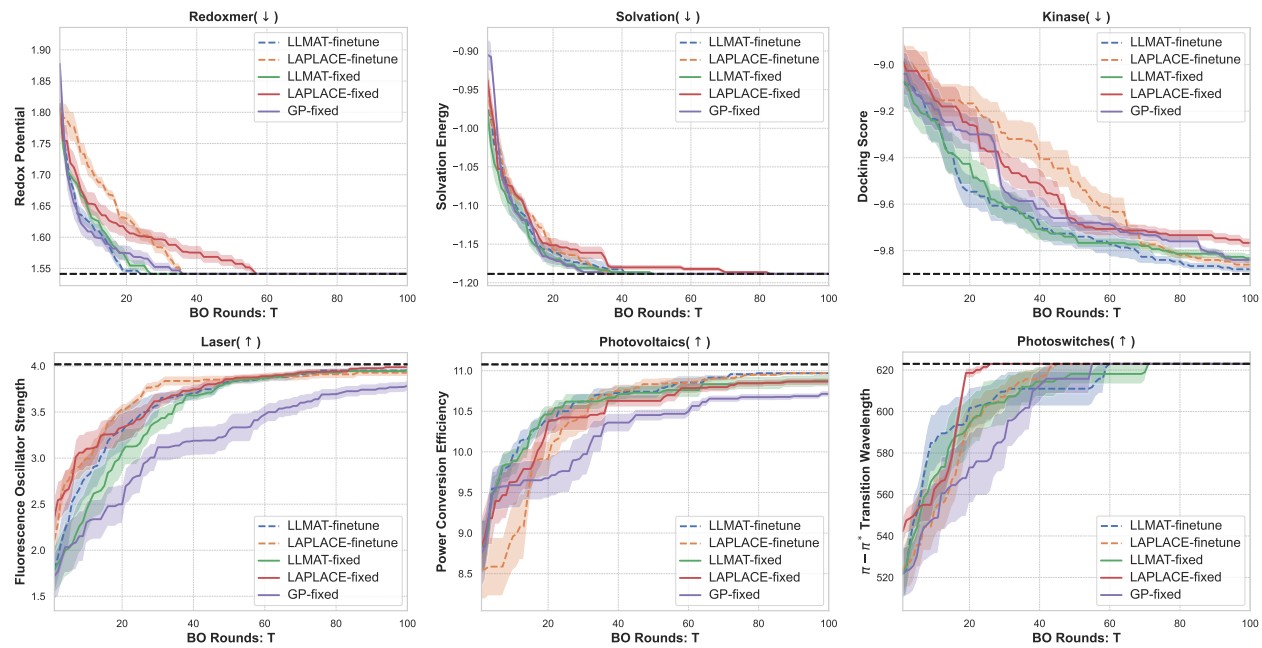

*Figure 23.* Historical optimums comparison on different chemistry datasets using Molformer.

### G.3.2. AVERAGE REGRETS

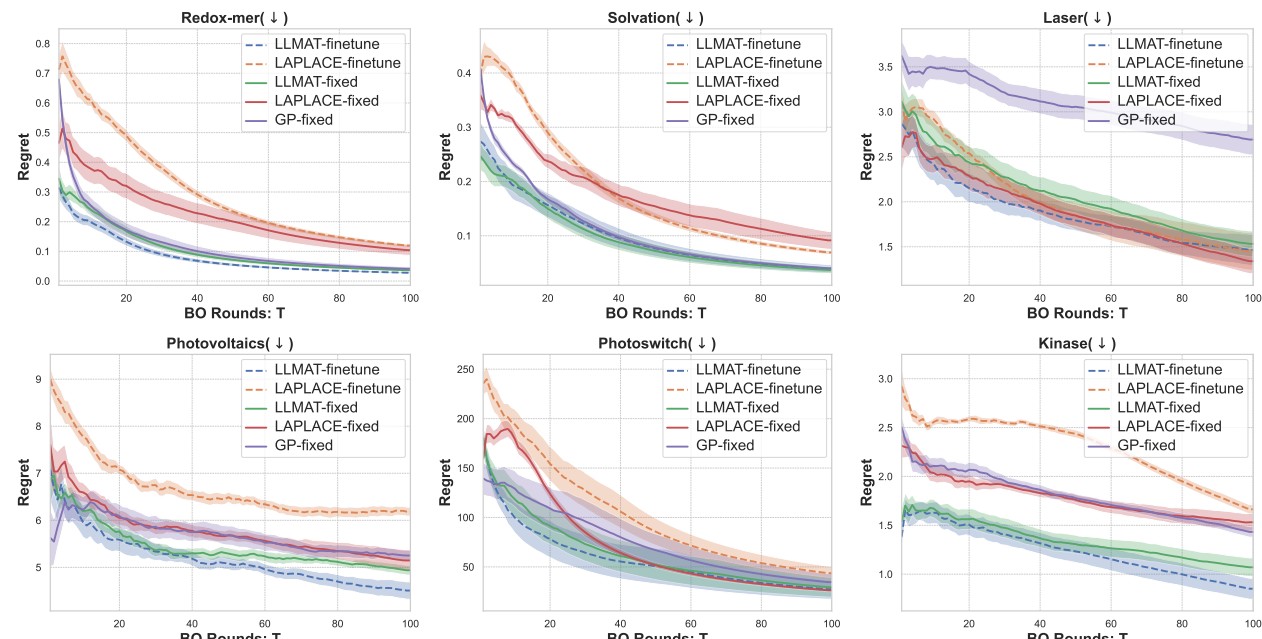

*Figure 24.* Regrets comparison on different chemistry datasets using Molformer.

### G.3.3. GAPs

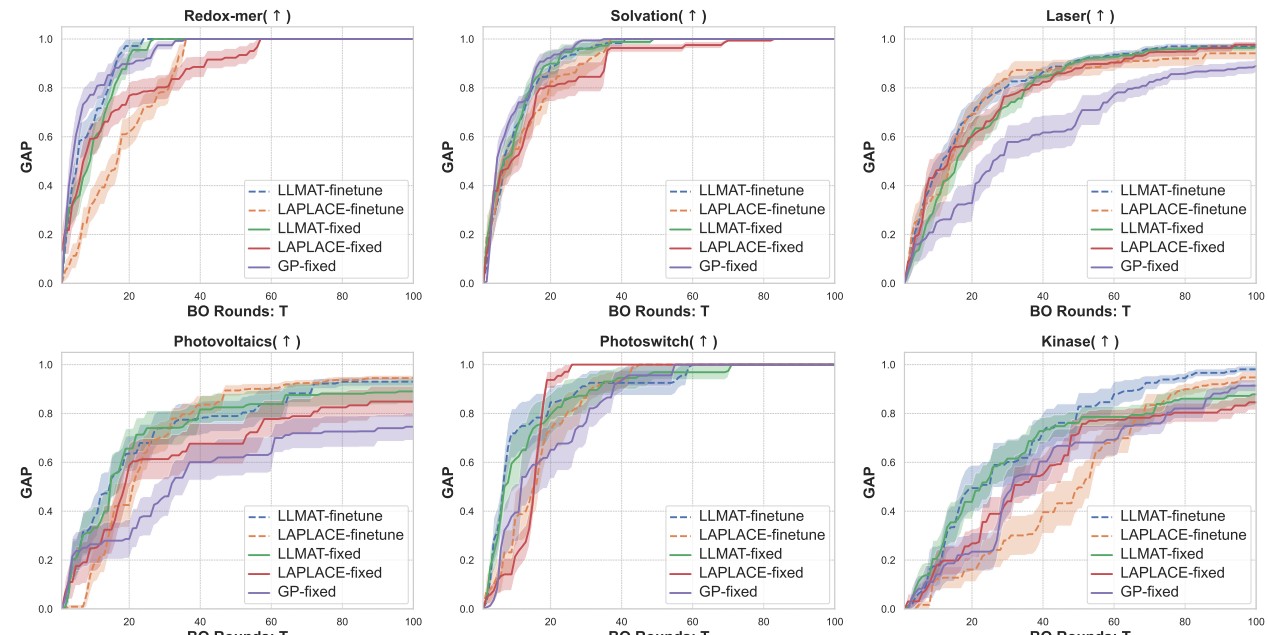

*Figure 25.* GAPs comparison on different chemistry datasets using Molformer

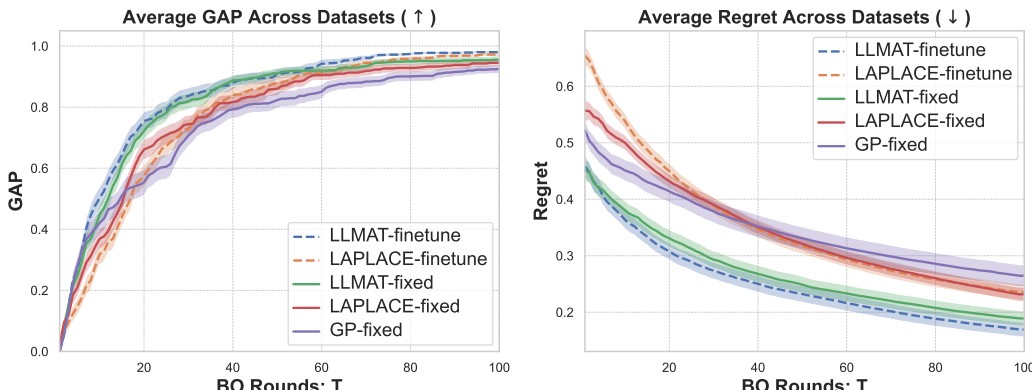

*Figure 26.* Average GAPs and regrets across all the datasets on finetuning and fixed Molformer.

## G.4. Clustering results

### G.4.1. K-MEANS CLUSTERING

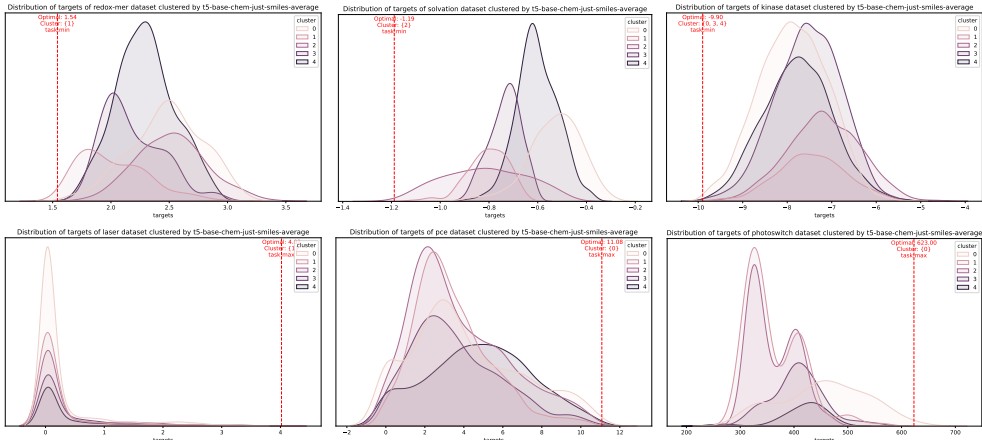

*Figure 27.* Distribution of property values for K-means clusters on features extracted from T5-Chem.

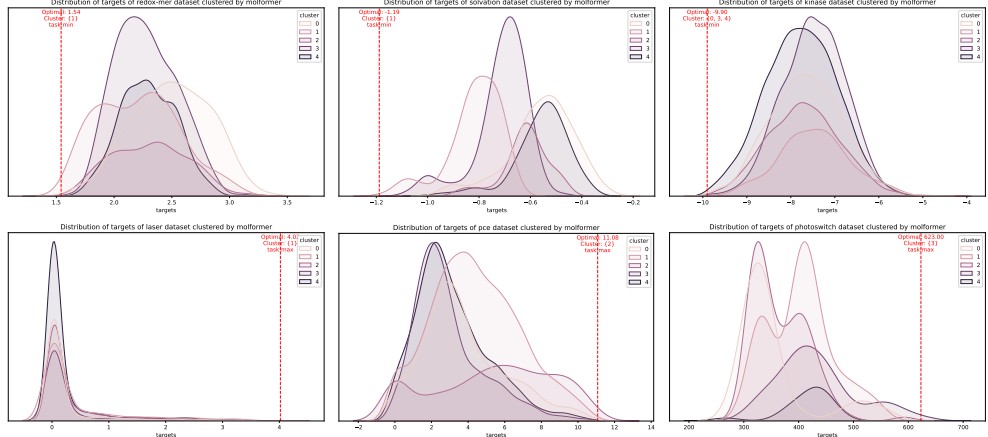

*Figure 28.* Distribution of property values for K-means clusters on features extracted from Molformer.

### G.4.2. LLM-BASED CLUSTERING

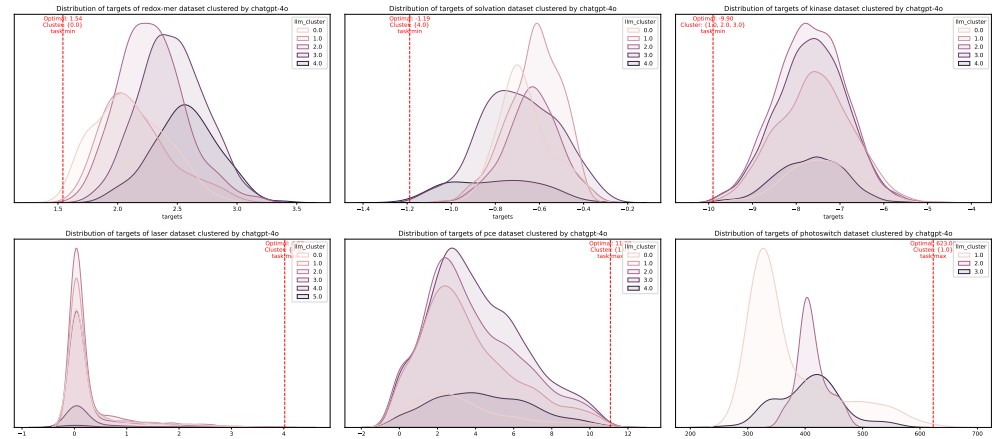

*Figure 29.* Distribution of property values for ChatGPT-based clustering results.

## G.5. Ablation on p-values

In this section, we provide additional details on how varying the p-values impacts our proposed algorithm in terms of the GAP metric, average regret, and prediction time per BO round for each dataset, using both K-means and LLM-based clustering. The results suggest that the LLM-based clustering approach is significantly effective in reducing prediction time while at the same time improving or maintaining the GAP and regret performance for Redoxmer, Solvation, and Photovoltaics, reflecting that ChatGPT-4o has informative knowledge on these datasets. This is consistent with its clustering visualization in Fig. 2, where the mean property values for clusters have an obvious difference. K-means clustering is effective for Redoxmer and Solvation in reducing prediction time without degrading the performance.

### G.5.1. GAPs' CHANGE W.R.T DIFFERENT P-VALUES

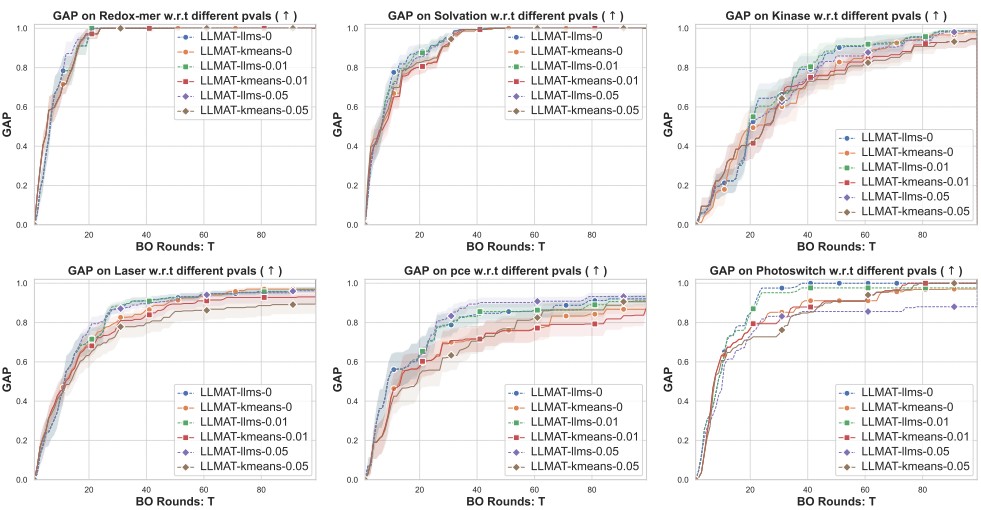

*Figure 30.* GAPs' change for different clustering approach and p-values using Molformer.

### G.5.2. REGRETS CHANGE W.R.T DIFFERENT P-VALUES

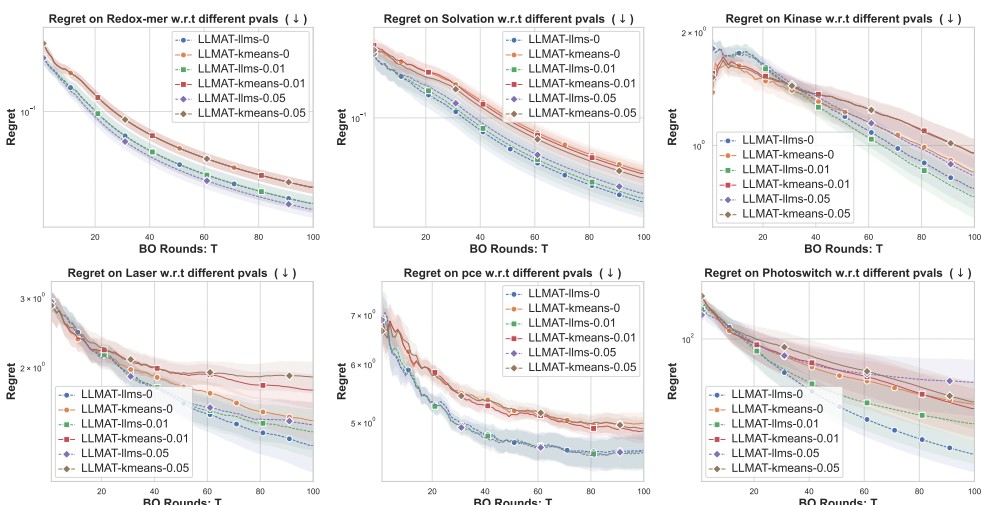

*Figure 31.* Regrets' change for different clustering approach and p-values with Molformer.

G.5.3. EFFECTS ON REDUCING PREDICTION TIME W.R.T DIFFERENT P-VALUES

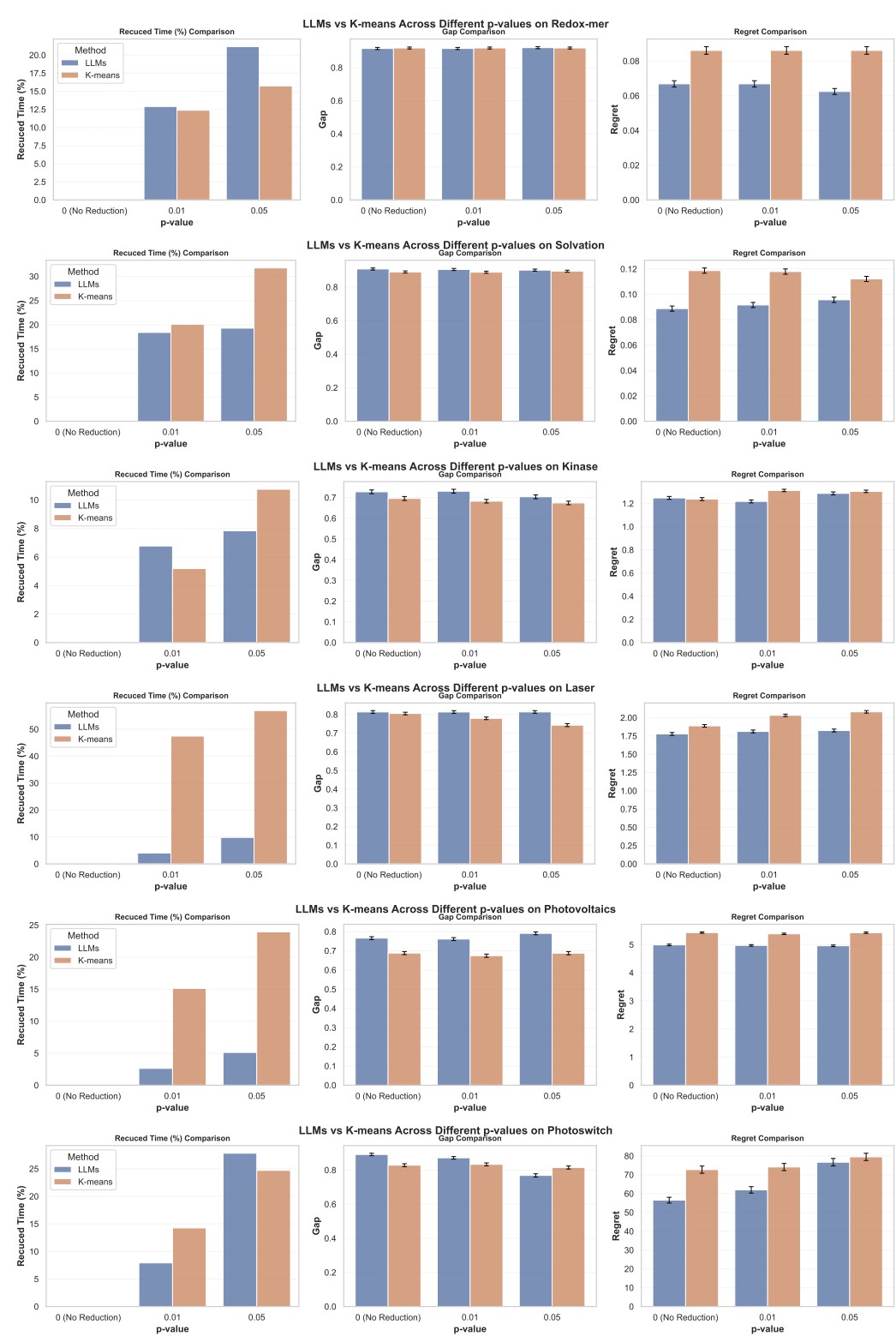

*Figure 32.* Prediction time reduction percentage, AUC of GAPs, and Regrets across all the datasets for different clustering approach and p-values.

### G.6. Toy Data

#### G.6.1. LEVY-1D DATA

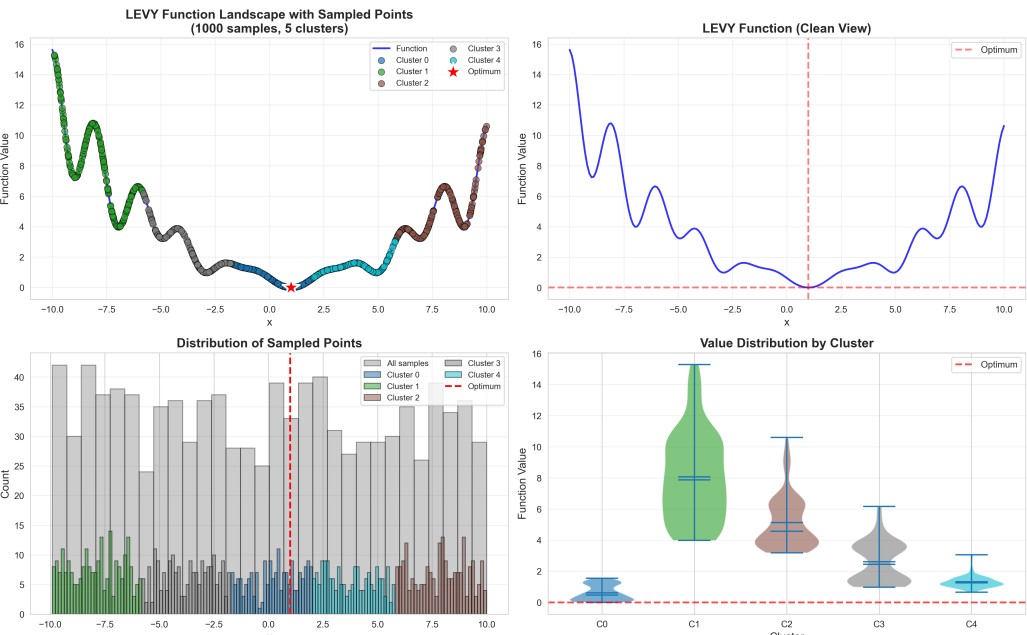

*Figure 33.* Illustration 1000 samples of Levy-1D function with K-means clustering.

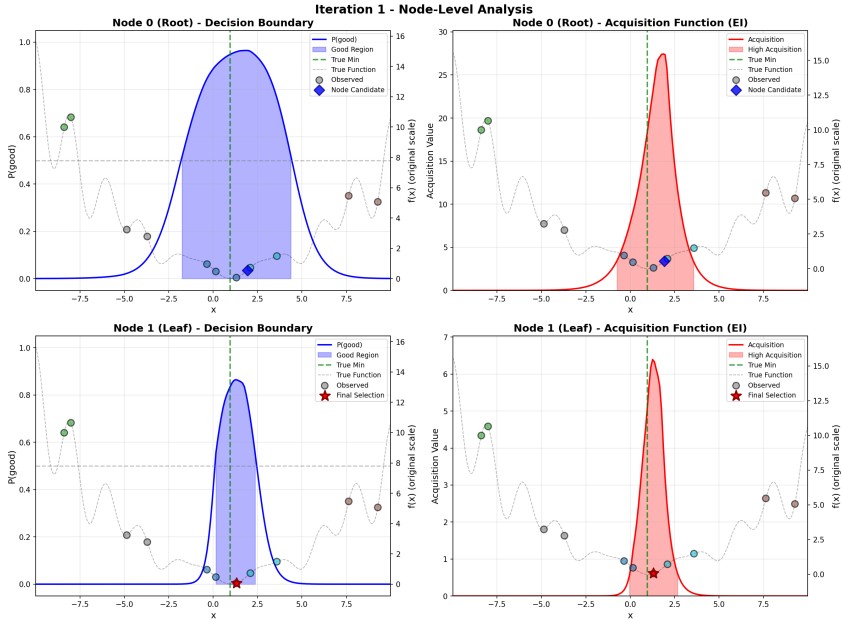

*Figure 34.* Refined AFs in LLMAT enable more efficient BO (Iteration 1).

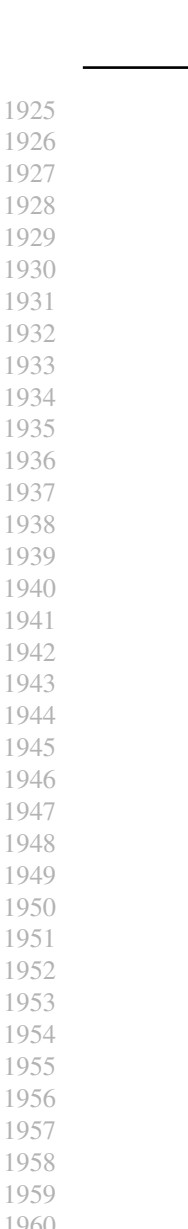

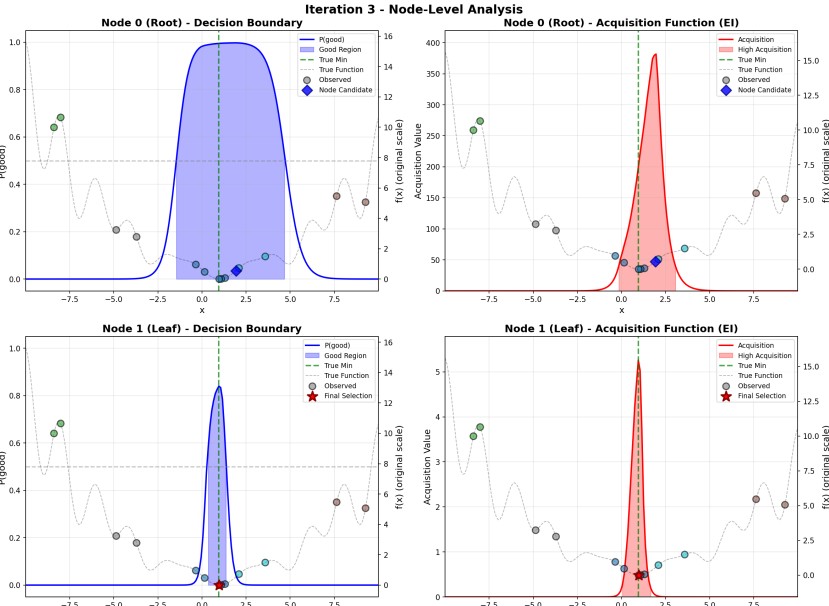

*Figure 35.* Refined AFs in LLMAT enable more efficient BO (Iteration 3).

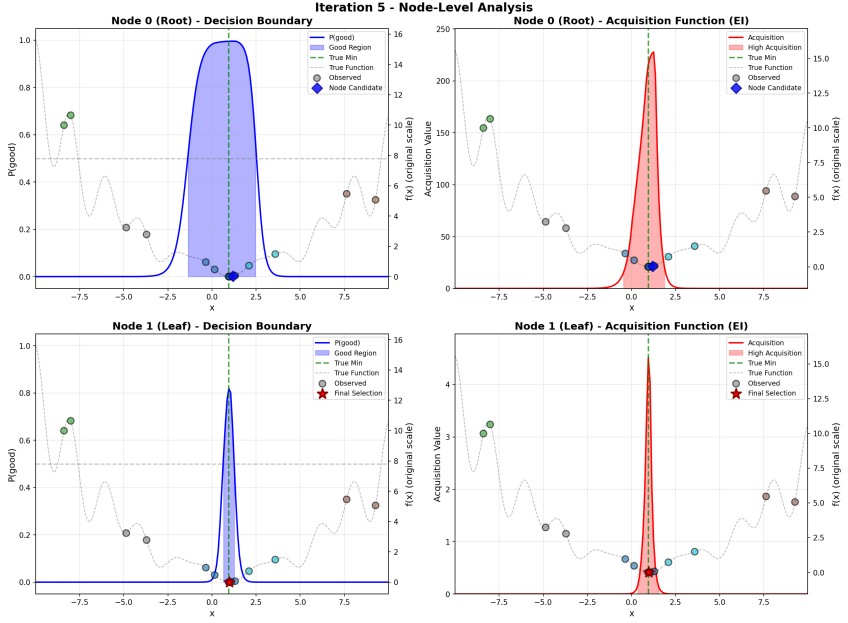

*Figure 36.* Refined AFs in LLMAT enable more efficient BO (Iteration 5).

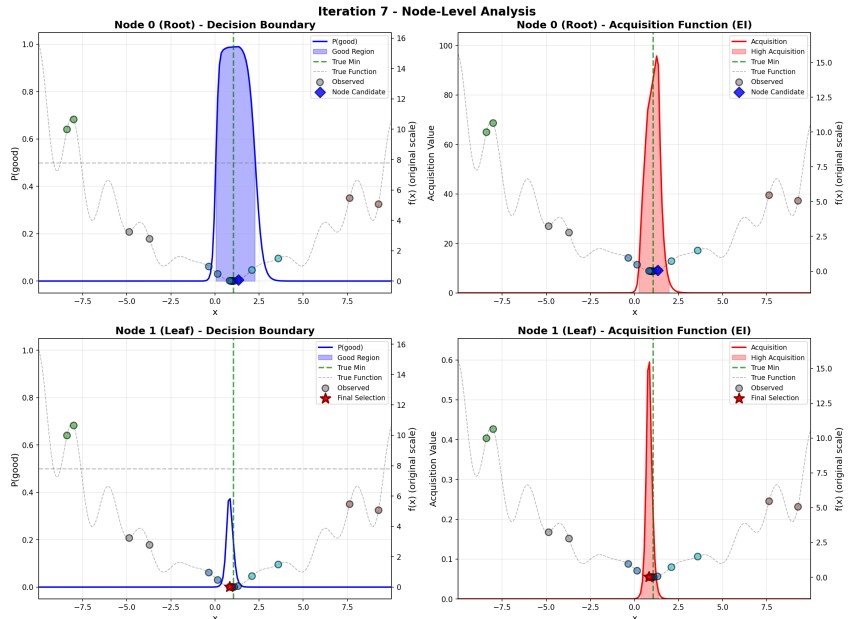

*Figure 37.* Refined AFs in LLMAT enable more efficient BO (Iteration 7).

### G.6.2. LEVY-10D DATA

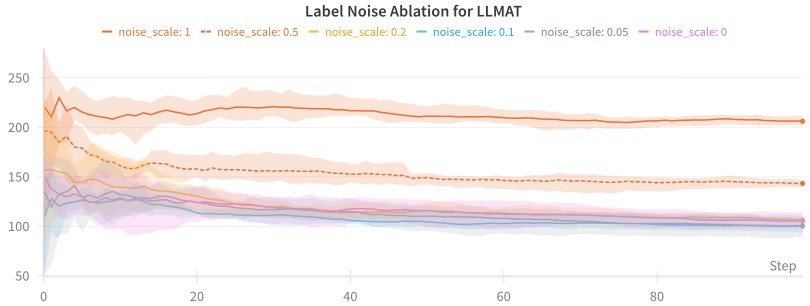

*Figure 38.* Illustration of Labal Noise's affects on the average regret of LLMAT.

### G.6.3. ABLATION OF $\gamma$

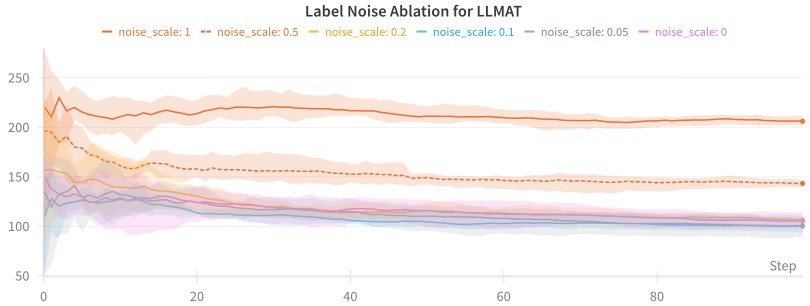

*Figure 39.* Illustration of LLMAT Performance Change w.r.t Different $\gamma$s.

### G.6.4. ADDITIONAL RESULTS FOR LLAM2-7B

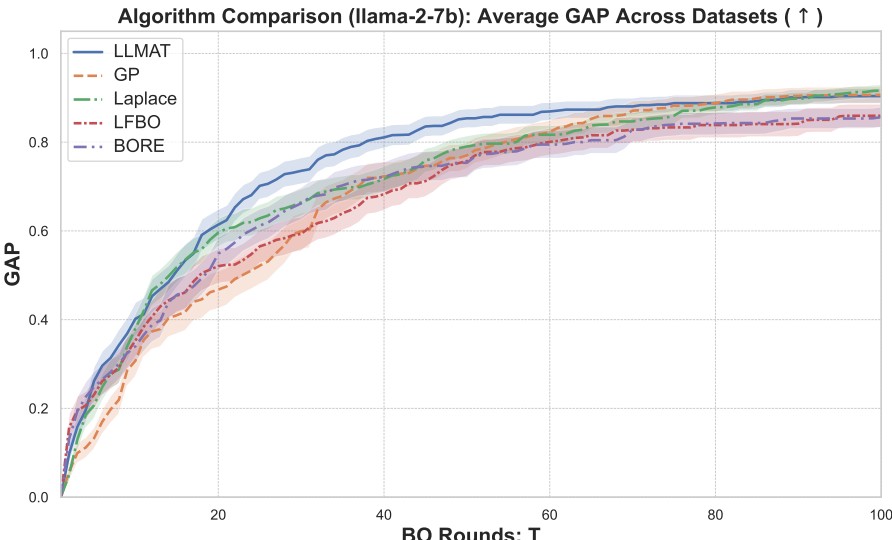

*Figure 40.* Illustration Performance Comparison of Various Algorithms on Llama2-7b across All Datasets.

## G.7. Multi-objective Optimization Extension

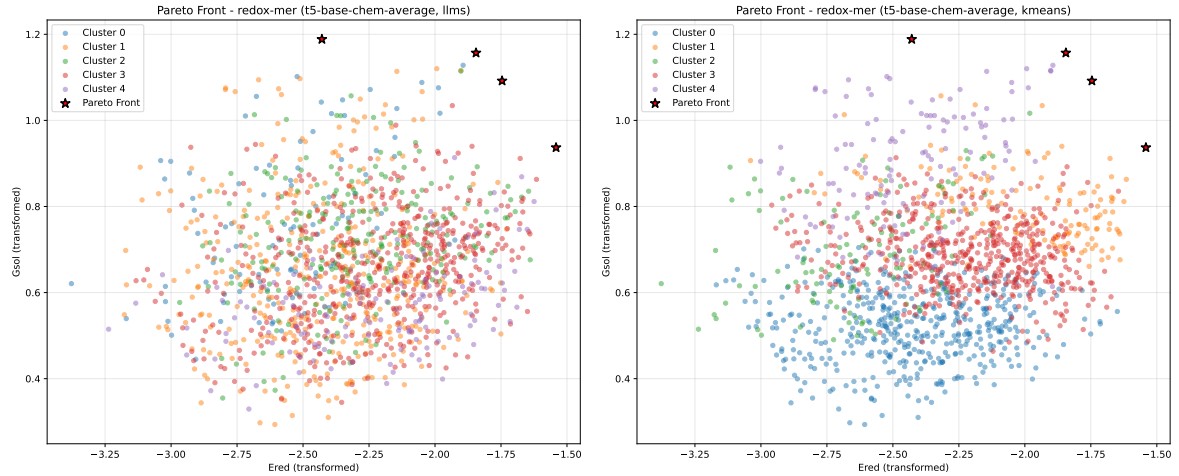

*Figure 41.* Illustration of Pareto fronts and LLMs and K-means clusters on Multi-Redox Data.

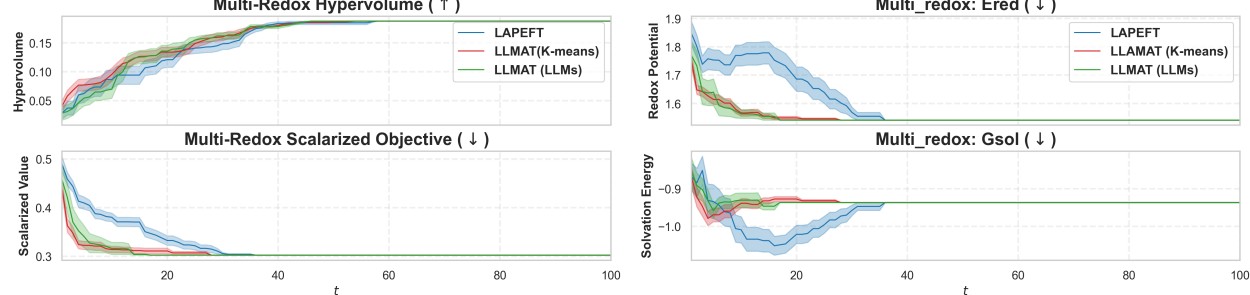

*Figure 42.* Multi-objective optimization for LAPEFT vs. LLMAT on the Redox-mer dataset: **Upper & Lower Left:** Hypervolume, scalarized objective. **Right:** Redox potential and solvation energy.

