# OpenReview forum: "Foundation Model Informed Acquisition Functions for Molecular Discovery"
_ICML.cc/2026/Conference — Submitted to ICML 2026_

### Official Review · Reviewer_kiji · 2026-03-11

**Soundness:** 2
**Presentation:** 2
**Significance:** 2
**Originality:** 2
**Overall Recommendation:** 2
**Confidence:** 3

**Summary:**

This paper presents a relatively complex Bayesian optimization workflow in which
- PEFT on transformers is used partition the candidate space
- Meta learning is used for the PEFT
- LLMs are used for increasing efficiency by clustering the design space and then only evaluate acquisition functions on "better" clusters.

**Compliance With Llm Reviewing Policy:**

Affirmed.

**Key Questions For Authors:**

I have no concrete questions but only the points mentioned above.

**Limitations:**

I feel there is room for more discussion of limitations - especially the assumptions the technique makes.

**Strengths And Weaknesses:**

Strengths: Bayesian Opt is an important problem and making it more data efficient is certainly valuable.

Presentation: I found the paper relatively hard to follow, as there are many things the authors change in the BO system. From reading it, it was very difficult to distill one key claim for the paper. There are many (small) changes that have been implemented for this paper, and this makes it difficult to extract a coherent message from it. And it is also unclear what parts of it could be generalized/transferred.
I feel if the authors were able to articulate one key contribution and framed the paper around it, it would dramatically improve the paper.  This would also increase my rating in "Presentation".

Weaknesses:

- The workflow is relatively complex, making it difficult to assess where and how LLMs can provide meaningful property ranking.
- Key assumptions appear quite strong: it is not clear that an LLM can perform meaningful ranking in many cases.
- The performance gains are marginal and not super big.
- Complex Bayesian optimization methods present a practical problem: often, there is no way to select good hyperparameters.
- It is unclear if any speed improvements are relevant: for meaningful problems, data generation will always be super expensive, which appears to be the fundamental bottleneck.

Technical issues/questions:
- PEFT with less data (which I understand the authors do) might make the resulting model worse in some cases, leading to a non-trivial complication of the entire system.
- Time comparisons are unclear and depend heavily on settings; in the worst case, they even depend on hardware.

Overall:
This is a relatively complex method and it is not clear if it really brings meaningful advances. The improvements are marginal and the assumptions are quite strong. This also contributes to my rating in "Soundness" and "Significance" - I do neither see a theoretical knowledge gain nor a practical impact in a real-world usecase of this technique.

---

> ### Author Rebuttal · Authors · 2026-03-30
>
> We sincerely thank the reviewer for the valuable and constructive feedback. Below, we address your concerns in detail.
>
> 🔴 **Additional plots are in https://anonymous.4open.science/r/ICML2026-Rebuttal-0E6A/Rebuttal.md.**
>
> > Presentation.
>
> Thank you for the suggestion. Our main contribution lies in the **algorithmic design (localized AF + tree partitioning)**, while LLM components serve as optional priors. Due to space limitations, we refer the reviewer to **our response to W1 of Reviewer xkYP** for additional details.
>
> > W1 Workflow is complex, difficult to assess LLMs in providing meaningful property ranking
>
> We thank the reviewer for this concern. Beyond Fig. 2 in submitted manuscript, we now include additional experiments across multiple LLM APIs to assess whether LLMs yield meaningful property rankings, comparing direct clustering (DC), property prediction (PP), and property-derived clustering (PC).
>
> | Model          | PP $R^2$↑ | DC HunAcc↑ | DC Spearman↑ | DC PairAcc↑ | PC HunAcc↑ | PC Spearman↑ | PC PairAcc↑ |
> | :------------- | :-----------: | :--------: | :----------: | :---------: | :--------: | :----------: | :---------: |
> | GPT-4o         |    -6.0016    |   0.3141   |    0.4943    |   0.7732    |   0.2459   |    0.1479    |   0.5795    |
> | Gemini-2.5-pro |   -87.9568    |   0.4471   |    0.8041    | **0.9084**  |   0.2232   |    0.2569    |   0.7163    |
> | Qwen3.5-Plus   |  **-0.8572**  | **0.4961** |  **0.8055**  |   0.8977    | **0.3689** |  **0.6510**  | **0.7845**  |
>
> * **Evaluation setup.** We evaluate property prediction using
> $R^2 = 1 - \frac{\sum_i (y_i - \hat{y}_i)^2}{\sum_i (y_i - \bar{y})^2}$, where $R^2 < 0$ indicates performance worse than a mean predictor. Ground-truth clusters are obtained via **quantile-based binning into five equal-frequency groups**. DC outputs labels directly, while for PC, predicted values are mapped to clusters using the same bin edges. We assess ranking via Spearman (rank correlation) and **PairAcc (pairwise ordering accuracy, aligned with BO objectives)**, and clustering via **Hungarian accuracy (HunAcc)**. Meaningful ranking is defined as consistency with ground-truth ordering under these metrics.
>
> This decomposition isolates where LLMs clustering contribute in the pipeline.
> * While property prediction is unreliable ($R^2 < 0$ for all models), DC achieves strong ranking performance (Spearman up to 0.80, PairAcc up to 0.91), **well above random (≈0 and 0.5)**.
> * Importantly, PC performs substantially worse than DC, indicating that LLMs capture **relative ordering directly rather than through numerical prediction**.
> * Overall, these results **empirically invalidate this concern**: **although LLMs fail at absolute prediction, they provide reliable ranking signals sufficient for BO**, as evidenced by strong PairAcc and consistent BO performance across LLM APIs (🔴see anonymous link).
>
> > W2 The performance gains are marginal and not super big.
>
> * While the improvements are not dramatic, they are **not marginal**, particularly in terms of regret. The gains are **consistent across datasets and feature models**, which is critical in BO. Moreover, compared to prior work (Kristiadi et al., 2024), our improvements match or exceed the typical gains reported.
>
> > W3 Complex Bayesian optimization often leads no way to select good hyperparameters.
>
> * We thank the reviewer for this concern. Although the method includes multiple components, they are **well disentangled** and introduce **minimal hyperparameter overhead**.
> * Most hyperparameters follow **standard/default settings** (e.g., UCB, LLM confidence), with $\gamma=0.5$ fixed for balanced partitions. Only the meta-learning and LoRA learning rates require light tuning within a small range.
> * We conduct **ablations over 15 random seeds** (vs. 5 in prior work), showing **robust and stable performance**.
> * We also use the same hyperparameters across feature models and APIs, demonstrating **robustness and practical ease of deployment**. We will clarify this in the revision.
>
> > W4 speed improvements V.S. data generation
>
> * If we understand correctly, “data generation” refers to constructing a candidate pool. While this can incur cost, **oracle evaluations remain a key bottleneck**, and BO aims to **minimize the number of evaluations**.
>
> * LLM-based clustering reduces a relatively small overhead per step, but this becomes **non-negligible at scale**, where large candidate pools require repeated screening. Improving efficiency here enhances overall practicality.
>
> > Q1 PEFT with less data might make the resulting model worse in some cases
>
> * Thanks for your question. This is the motivation for introducing a meta-learning module for improving the stability of the algorithm.
>
> > Q2 Time comparisons are unclear and depend heavily on settings
> * As we claimed in the computational resource section: "We ensured that, for each dataset, all algorithms were tested on the same type of CPU and GPU."

---

> > ### Author Rebuttal · Reviewer_kiji · 2026-04-03
> >
> > My point about "W4 speed improvements V.S. data generation" and "W3 Complex Bayesian optimization often leads no way to select good hyperparameters." are related.
> >
> > In practical problems, for instance in the chemical sciences, the generation of experimental data is slow. Thus, algorithmic advances in making an evaluation of the algorithm faster do not solve the main bottleneck there.
> > For this real bottleneck, it would be important that the algorithm is robust and that performance can be achieved irrespective of the detailed settings for different datasets.
> >
> > If this were demonstrated for practically relevant datasets, I would be happy to increase my score.

---

> > > ### Author Response · Authors · 2026-04-05
> > >
> > > We thank the reviewer for confirming that most concerns have been resolved and for the follow-up. Below, we address the remaining points. We hope this clarifies the concerns on practical relevance and robustness, and would appreciate it being considered in the final assessment.
> > >
> > > > **...making an evaluation of the algorithm faster do not solve the main bottleneck there.**
> > >
> > > * We agree that **oracle evaluations (e.g., DFT or generation of experimental data) are the primary bottleneck** in molecular discovery. This is precisely why improving BO performance is critical: its goal is to reduce the number of expensive evaluations required.
> > > * Our contribution directly targets this bottleneck by achieving better solutions with fewer BO rounds, which translates into reduced DFT or experimental cost. This is consistently demonstrated in our experiments.
> > > * Efficiency gains on **evaluation of the algorithm** are secondary and optional. They **complement, but do not replace**, the core contribution of improved BO performance.
> > > * That said, it remains important in realistic settings, especially when integrating iterative BO with foundation models. Even before reaching the oracle stage, **candidate selection itself can become a computational bottleneck at scale**.
> > >     - For example, selecting one molecule from 10,000 candidates per BO round takes about 4 minutes for **LAPLACE** on a single GPU. Scaling to an extremely large candidate pool such as ZINC + PubChem (~1.1B molecules) would require ~277 days per round, whereas our method (LLMAT) reduces this to ~27.7 days. This is still only for a single BO round.
> > >     - In comparison, typical DFT evaluations take:
> > >         * Small molecules (10–50 atoms): minutes to hours
> > >         * Medium molecules (50–200 atoms): hours to days
> > >         * Large systems (>200 atoms): days to weeks
> > >     - Thus, improving computational efficiency at the selection stage is crucial for large-scale candidate pools. While parallelization can help, algorithmic efficiency directly reduces total GPU hours.
> > >
> > > > **It would be important that the algorithm is robust and that performance can be achieved irrespective of the detailed settings for different datasets.**
> > >
> > >    * We respectfully note that the expectation of performance being invariant to dataset-specific settings is **not aligned with standard machine learning principles**.
> > >    * In particular, the **No Free Lunch Theorem** (Wolpert & Macready, 1997) implies that performance **inevitably depends on task-specific inductive biases**  (Shalev-Shwartz & Ben-David, 2014).
> > >    * In BO, such dependence is fundamental: acquisition functions, model assumptions, and hyperparameters govern the exploration–exploitation trade-off, **making invariance unrealistic**. Robustness should instead mean consistent performance under shared, reasonable settings.
> > >    * In practice, our method demonstrates strong robustness with **largely shared hyperparameters** across diverse datasets.
> > >    * Moreover, in realistic settings with expensive evaluations, a small number of initial BO rounds can be used to select hyperparameters, after which a stable configuration is reused for subsequent rounds.
> > >
> > > > **If this were demonstrated for practically relevant datasets, I would be happy to increase my score.**
> > >
> > > We clarify that our experiments are conducted on **practically relevant** benchmarks **widely used** in molecular design.
> > >
> > > * | Benchmark                                        | Year | Property                    | Label source                 |
> > > | ------------------------------------------------ | ---- | --------------------------- | ---------------------------- |
> > > | Redoxmers (Agarwal et al., 2021)                 | 2021 | Redox potential             | DFT simulation               |
> > > | Flow battery electrolytes (Agarwal et al., 2021) | 2021 | Solvation energy            | Physics-based simulation     |
> > > | Kinase inhibitors (Graff et al., 2021)           | 2021 | Docking score               | Docking simulator            |
> > > | Laser materials (Strieth-Kalthoff et al., 2024)  | 2024 | Oscillator strength         | TD-DFT simulation            |
> > > | Photovoltaics (Lopez et al., 2016)               | 2016 | Power conversion efficiency | Simulation / surrogate model |
> > > | Photoswitches (Griffiths et al., 2022)           | 2022 | Transition wavelength       | TD-DFT simulation            |
> > >
> > >  * These datasets correspond to **real chemistry objectives** (e.g., redox potential, solvation energy, docking score), with labels obtained from **physics-based simulators** such as DFT, TD-DFT, or docking engines.
> > >  * Such benchmarks are the **standard evaluation protocol in BO for molecular discovery**, as they provide reproducible and controlled approximations of expensive real-world oracles.
> > >  * While fully interactive wet-lab validation is desirable, it is **beyond the scope of this work** and not standard in prior literature, which similarly relies on simulation-based benchmarks.

---

### Official Review · Reviewer_xkYP · 2026-03-12

**Soundness:** 3
**Presentation:** 2
**Significance:** 2
**Originality:** 3
**Overall Recommendation:** 4
**Confidence:** 4

**Summary:**

The paper proposed a molecular BO method, which leverages tree-based local acquisition search on FM embeddings, with optional LLM clustering to reduce cost.

**Compliance With Llm Reviewing Policy:**

Affirmed.

**Final Justification:**

Thanks authors for the rebuttal, I have no more questions, and will increase the score to 4.

**Key Questions For Authors:**

- The core contribution feels somewhat diffuse. Can the authors clarify what they view as the single main technical novelty: the tree partitioning itself, the local AF design, the MCTS traversal, or the LLM-based clustering?

- The LLM-based clustering module looks useful for efficiency, but its contribution to optimization quality seems modest. How essential is this module to the overall method, and would the paper be nearly as strong without it?

- The clustering approach is described by the authors themselves as prototypical and based on simple GPT-4o prompts. How robust is this component to prompt choice, model choice, and API variability, and how much should readers view it as a core contribution versus a practical heuristic?

**Limitations:**

I don't know

**Strengths And Weaknesses:**

Strengths:

- The paper tackles a meaningful problem: Bayesian optimization for molecular discovery with very limited labels and a large discrete candidate pool. The motivation is practical and well aligned with the difficulty of fitting a strong global surrogate in this setting.

- The overall method is sound. The tree partitioning, local AF learning, and MCTS/UCB-style traversal fit together naturally as a localized alternative to global BO.

- The empirical study is reasonably broad. It includes six chemistry datasets, multiple feature backbones, and several BO baselines, which gives the method a fairly solid evaluation scope.

Weaknesses:

- The main limitation is that the core novelty feels somewhat compositional rather than a new framework. The method combines tree partitioning, classifier-based local AF learning, MCTS/UCB traversal, meta-learning, and LLM clustering, but it is less clear which part is the main conceptual advance.

- The LLM-based clustering component feels somewhat secondary and under-validated. It looks more like a practical filtering heuristic than a deeply integrated core contribution, and the evidence for its necessity is not fully convincing.

- The paper is generally understandable, but the number of compositional components makes the contribution feel slightly diffuse. As a result, the central message can be harder to isolate than in papers with one cleaner technical idea.

---

> ### Author Rebuttal · Authors · 2026-03-31
>
> We sincerely thank the reviewer for the valuable and constructive feedback. Below, we address your concerns in detail.
>
> 🔴 **Additional plots are provided in https://anonymous.4open.science/r/ICML2026-Rebuttal-0E6A/Rebuttal.md.**
>
> > W1 Core novelty seems to be compositional and less clear which part is the main conceptual advance. & W3 Insufficient central message for clearer technical idea & Q1 Can the authors clarify what they view as the single main technical novelty?
>
> We thank the reviewer for this important feedback and agree that the presentation could better emphasize a single conceptual advance. Our results show that **performance is primarily driven by algorithmic design (localized AF + tree partitioning), while LLMs act only as optional priors.**
>
> * Our main contribution is a **novel discrete BO framework based on tree-structured partitioning with shared classifiers**, which jointly (i) constructs a hierarchical partition of the search space and (ii) learns localized acquisition functions (AFs) within each partition. This formulation enables efficient candidate selection and AF optimization via Monte Carlo Tree Search, **while avoiding explicit surrogate modeling**. As a result, it offers **broader applicability to BO settings involving recent advances in LLMs and foundation models**. Overall, this constitutes the primary technical advance of the paper.
>
> * The other components are not independent additions, but **natural instantiations within this framework**. In particular, meta-learning improves the robustness of the shared classifiers in low-data regimes, while LLMs can be incorporated as **prior signals to guide partitioning and candidate suggestion**.
>
> * Our results show that localized AF learning via tree partitioning is the primary driver of performance, with LLMs serving as coarse structural priors. Even without LLM-based clustering, the method remains strong (at increased computational cost), confirming that the core contribution is algorithmic design.
>
> We will revise the paper to better highlight this central idea.
>
>
> > W2 LLM-based clustering somewhat under-validated & Q2 How essential is the LLM-based clustering approach?
>
> We thank the reviewer for this insightful comment and agree that the role of LLM-based clustering should be clarified.
> * **LLM-based clustering is not core to our method**, but a modular component for reducing the cost of evaluating AFs over large candidate sets. The method remains strong without it, confirming that the main contribution is algorithmic (Fig. 9).
>
> * Its benefit **scales with LLM quality**: stronger models can filter irrelevant regions and improve both efficiency and performance (see Fig. 32 in Appendix), while weaker ones mainly provide **computational savings (10–25%)**.
>
> * Additional experiments with stronger LLMs (Gemini-2.5-pro, Qwen3.5-Plus) support this trend but do not change the overall conclusion (**🔴See Fig. 2–3 in Additional plots**).
>
> * Overall, we view it as a **plug-in component that improves efficiency and adapts to model quality**, rather than a necessary ingredient.
>
> > Q3 How robust is LLM clustering to prompt choice, model choice, and API variability?
>
> We thank the reviewer for this question and clarify the robustness and role of the LLM-based clustering component.
>
> * **Prompt robustness.** As shown in Fig. 9 (submitted manuscript), clustering performance (measured by GAP) remains largely stable across different prompts. This robustness stems from the statistical test used in cluster selection, which incorporates observed data and reduces sensitivity to prompt variations.
>
> * **Model / API robustness.** We have added experiments on more recent LLMs (Gemini-2.5-pro and Qwen3.5-Plus) on the redox-mer dataset. Despite differences in absolute clustering and ranking quality, BO performance remains consistent (🔴Fig. 2, anonymous link), with stronger models yielding better alignment with ranking metrics.
>
> * | Model           | R² ↑     | HunAcc ↑ | Spearman ↑ | PairAcc ↑ |
> |-----------------|----------|----------|------------|-----------|
> | GPT-4o          | -6.0016  | 0.3141   | 0.4943     | 0.7732    |
> | Gemini-2.5-pro  | -87.9568 | 0.4471   | 0.8041     | **0.9084**|
> | Qwen3.5-Plus    | **-0.8572** | **0.4961** | **0.8055** | 0.8977    |
>
> * **Role of the component.** We emphasize that LLM-based clustering is best viewed as a **practical, modular heuristic** that provides prior-guided partitioning. It is **a secondary contribution**, and the method remains strong without it. Its main benefit is improved efficiency and slightly better early-stage guidance, while overall performance is primarily driven by the algorithmic framework.
>
> Overall, these results indicate that the component is **robust to prompt, feature models, and API choices**, and should be viewed as an optional, plug-in module rather than a critical dependency. We will clarify this positioning in the revision.

---

> > ### Author Rebuttal · Reviewer_xkYP · 2026-04-03
> >
> > Thanks authors for the rebuttal, I have no more questions, and will increase the score to 4.

---

> > > ### Author Response · Authors · 2026-04-05
> > >
> > > We thank the reviewer for confirming that the concerns have been fully addressed. We are glad that the clarifications were helpful. We also sincerely appreciate your thoughtful feedback and careful engagement, which have helped improve the clarity and quality of the paper.

---

### Official Review · Reviewer_E7Vm · 2026-03-13

**Soundness:** 3
**Presentation:** 2
**Significance:** 3
**Originality:** 3
**Overall Recommendation:** 4
**Confidence:** 3

**Summary:**

The paper proposes a likelihood-free BO framework named LLMAT without the need of explicit modeling of a surrogate model. Instead, the proposed model uses priors from general LLMs and chemistry foundation models. Furthermore, it partitions the molecular candidate space into a tree structure for efficient candidate selection via Monte Carlo Tree Search (MCTS). From the author's experimental results, LLMAT shows better performance than baseline approaches.

**Compliance With Llm Reviewing Policy:**

Affirmed.

**Key Questions For Authors:**

Please refer to weaknesses section.

**Limitations:**

The authors have adequately discussed the limitations and potential negative societal impact.

**Strengths And Weaknesses:**

## Strengths
- The paper is easy to read and well-written.
- The proposed LLMAT shows better optimization results compared to other baselines such as LAPLACE.
- I think that the proposed method is novel and interesting. Tree-based search is like a trust-region-based local search, but the trust-region-based local search typically depends on distance metrics, which are hard to define in language domain. From this perspective, the tree-based search makes sense to me.

## Weaknesses
- It would be better if the paper discussed why meta-learning is necessary in LLAMAT. I think that updating both $\theta$ and $\omega$ shows better performance than meta-leraning-based training mechanism.
- Could you provide the experimental results based on more recent LLMs (e.g., Qwen3) to demonstrate the generalizability of LLMAT?

---

> ### Author Rebuttal · Authors · 2026-03-31
>
> We sincerely thank the reviewer for the valuable and constructive feedback. Below, we address your concerns in detail.
>
> 🔴 **Additional plots are provided in https://anonymous.4open.science/r/ICML2026-Rebuttal-0E6A/Rebuttal.md.**
>
> > **W1 Why meta-learning is necessary in LLMAT?**
>
>
> We thank the reviewer for this insightful comment.
>
> Meta-learning is particularly important in our setting, where BO operates in a **low-data regime (10 initial points)** and only a limited number of evaluations are available to initialize the model. Directly updating all components from scratch in this regime can lead to **unstable optimization and poor generalization**.
>
> As shown in the upper plot of Fig. 9, meta-learning yields a **substantial performance improvement** over standard joint training. This is because it learns a **shared prior across partition trees**, which regularizes the updates and prevents large, noisy parameter shifts when data is scarce. Such inductive bias is crucial for constructing reliable acquisition functions from very limited observations.
>
> In contrast, without meta-learning, the model relies solely on sparse observations, making it more prone to overfitting and unstable updates. Therefore, meta-learning is not only beneficial but **essential for stabilizing training and improving performance in the low-data BO setting**.
>
> -------------------
> > **W2 Experiments on more recent LLMs like Qwen3.**
>
> * We have added experiments on the redox-mer dataset for Gemini-2.5-pro and Qwen3.5-Plus. BO performance (**🔴Fig. 2–3, Additional plots**) is consistent across APIs, with more advanced models (Gemini and Qwen) achieving better results.
>
> * We further evaluate their clustering and prediction performance, and find that BO performance aligns with stronger ranking metrics (PairAcc), where Gemini outperforms.
> | Model          | $R^2$↑ | HunAcc↑ | Spearman↑ | PairAcc↑ |
> | :-------- | :----: | :-----: | :-------: | :------: |
> | GPT-4o         | -6.00  | 0.3141  | 0.4943    | 0.7732   |
> | Gemini-2.5-pro | -87.96 | 0.4471  | 0.8041    | **0.9084** |
> | Qwen3.5-Plus   | **-0.86** | **0.4961** | **0.8055** | 0.8977   |
>
> * **Evaluation setup.** We evaluate property prediction using
> $R^2 = 1 - \frac{\sum_i (y_i - \hat{y}_i)^2}{\sum_i (y_i - \bar{y})^2}$, where $R^2 < 0$ indicates performance worse than a mean predictor. Ground-truth clusters are obtained via *quantile-based binning into five equal-frequency groups*. Ranking is evaluated using Spearman and **PairAcc** (pairwise ordering accuracy), and clustering quality using Hungarian accuracy (**HunAcc**).

---

> > ### Author Rebuttal · Reviewer_E7Vm · 2026-04-04
> >
> > My concerns have been adequately addressed. After reading the reviews of other reviewer and their corresponding rebuttals, I maintain my score.

---

> > > ### Author Response · Authors · 2026-04-05
> > >
> > > We thank the reviewer for confirming that the concerns have been adequately addressed. We are glad that the clarifications were helpful. We also sincerely appreciate your thoughtful feedback and careful engagement, which have helped improve the clarity and quality of the paper.

---

### Official Review · Reviewer_2NEK · 2026-03-13

**Soundness:** 2
**Presentation:** 2
**Significance:** 3
**Originality:** 3
**Overall Recommendation:** 4
**Confidence:** 3

**Summary:**

The authors propose LLM-guided acquisition trees as an approach for sample efficient molecular optimization, which operates by (i) featurizing molecules in SMILES form with pre-trained foundation LLMs, and (ii) partitions the context dataset via MCTS and fits a logistic regression based acquisition function on these partitions. Through extensive experimentation, the authors demonstrate their proposed method is able to optimize molecules against computational endpoints and with competitive performance relative to other non-LLM based Bayesian optimization approaches.

**Compliance With Llm Reviewing Policy:**

Affirmed.

**Final Justification:**

This paper presents a timely and well executed framework for integrating large language models into Bayesian optimization, offering a novel approach to molecular discovery supported by an extensive experimental section and supplementary analyses. While the technical contribution is sound, the presentation would benefit from a more focused distillation/synthesis of the extensive supplementary data to more clearly highlight the primary takeaways and key results. Ultimately, the work represents a solid contribution to this problem area, and although the scope of the application is somewhat narrow, I recommend its acceptance.

**Key Questions For Authors:**

Can you clarify exactly how the molecules are encoded into features that are used for fitting the acquisition functions? Is it simply encoding the SMILES with the LM in question and taking a sequence-level embedding (e.g., the [CLS] output embedding)? Or are you currently (or have you considered) encoding the SMILES along with a prompt that describes the task in question? Essentially, I am wondering if there is any lift in performance from encoding the task + SMILES versus just the SMILES for purposes of fitting the acquisition functions.

**Limitations:**

Yes

**Strengths And Weaknesses:**

Strengths:
- Acquisition function strategy is both interesting and novel, has the advantage that it avoids explicit surrogate modeling and may have more general utility to applications of BO to LLMs.
- The authors have included an extensive supplemental materials section with many relevant experiments demonstrating LLMAT's performance. I found many of these to highlight the method positively.

Weaknesses:
- It is unclear how significant of an issue data leakage is in the evaluation of LLMAT on the reported tasks in comparison to the non-LLM based BO approaches. There is an argument to be made that the effective number of context examples these models operate on is larger as the features are themselves a function of the data the LLM was trained on, and some of that data may coincide with what is reflected in the evaluation tasks.
- It appears that all experiments in the paper are limited to chatgpt-4o. This seems like an important component to ablate.
- The writing and presentation of key information can be improved. The authors should put more effort into distilling some of the main takeaways from the thorough experimentation that is provided in the supplement, specifically section G.

---

> ### Author Rebuttal · Authors · 2026-03-31
>
> We thank the reviewer for the valuable and constructive feedback. Below we address your concerns in detail.
>
> 🔴 **Additional plots are provided in https://anonymous.4open.science/r/ICML2026-Rebuttal-0E6A/Rebuttal.md.**
>
> >W1: Unknown data leakage in evaluation compared to non-LLM BO
>
> Thanks for raising this important concern, which is broadly relevant to LLM-based BO methods. We consider two sources of LLM priors in our framework and provide empirical evidence below that data leakage is unlikely to explain our results for either case.
>
> * LLM representation.
>     - Under the same GP, fingerprint features perform comparably to LLM features (Fig. 11), inconsistent with a memorization advantage. Moreover, all methods improve gradually over BO iterations rather than converging early, further ruling out retrieval of memorized solutions. Thus, LLM representations do not act as a shortcut, but provide a generic prior whose effectiveness depends on algorithmic design, as reflected in the consistent gains of our method across datasets (Fig. 5).
>
> * LLM clustering.
>     - If leakage dominated, we would expect accurate prediction. However, property prediction is highly unreliable (see following Table, all models have $R^2 < 0$), while ranking remains strong (Spearman up to 0.80, PairAcc up to 0.91). This pattern is inconsistent with memorization and instead suggests that LLMs capture coarse chemical heuristics inducing relative ordering.
>
> * Conclusion: LLMs do not provide memorized solutions but rather generalizable priors. We will clarify this in the revision.
>
> >W2: No ablation for the LLM clustering API
>
> * We have added experiments on the redox-mer dataset for Gemini-2.5-pro and Qwen3.5-Plus. BO performance (🔴**Fig. 2–3, Additional plots**) is consistent across APIs, with more advanced models (Gemini and Qwen) achieving better results.
>
> * We further evaluate their clustering and prediction performance, and find that BO performance aligns with stronger ranking metrics (PairAcc), where Gemini outperforms.
> | Model          | $R^2$↑ | HunAcc↑ | Spearman↑ | PairAcc↑ |
> | :-------- | :----: | :-----: | :-------: | :------: |
> | GPT-4o         | -6.00  | 0.3141  | 0.4943    | 0.7732   |
> | Gemini-2.5-pro | -87.96 | 0.4471  | 0.8041    | **0.9084** |
> | Qwen3.5-Plus   | **-0.86** | **0.4961** | **0.8055** | 0.8977   |
>
> * **Evaluation setup.** We evaluate property prediction using
> $R^2 = 1 - \frac{\sum_i (y_i - \hat{y}_i)^2}{\sum_i (y_i - \bar{y})^2}$, where $R^2 < 0$ indicates performance worse than a mean predictor. Ground-truth clusters are obtained via *quantile-based binning into five equal-frequency groups*. Ranking is evaluated using Spearman and **PairAcc** (pairwise ordering accuracy), and clustering quality using Hungarian accuracy (**HunAcc**).
>
> >W3: Lack of distilled takeaways
>
> Thanks for the helpful suggestion. Due to space constraints, we did not explicitly distill the key takeaways; we will add the following in the revision:
>
> Our results suggest **how** to effectively use LLM for BO. In particular, we show that directly using LLMs as predictors for complex chemistry task is suboptimal, while a **prior-guided, algorithm-centric integration** leads to substantially better performance (Table; Fig. 5, 11, 15–17, 23–26).
> * The main performance gains stem from localized acquisition function (AF) learning via tree partitioning: increasing tree depth improves GAP and regret, while LFBO (no partitioning) performs significantly worse (Fig. 7, 9).
> * Meta-learning is critical in low-data regimes, improving stability and final performance (Fig. 9).
> * LLMs serve as coarse structural priors, not accurate predictors: while their numeric predictions are noisy, their ranking signal is sufficient to guide clustering and restrict the search space (Table; Fig. 1, anonymous link).
> * LLM-based clustering improves computational efficiency (10–25% reduction), early-stage convergence (Fig. 9) and is robust to prompt and API variations (Fig. 8, 9, 30–32).
>
> Overall, **algorithmic design (local AF + tree partitioning)** is the dominant factor, with LLMs providing a lightweight, robust prior.
>
> > Q1: Molecule encoding and effect of prompting
>
> * In our implementation, molecules are encoded by passing SMILES strings through the LLM and extracting a **sequence-level embedding via masked mean pooling over token representations**.
> * Prior work (e.g., Kristiadi et al., 2024, _“A Sober Look at LLMs for BO”_, Sec. 4.3, Fig. 6 & 11) has shown that prompt-based representations can influence BO performance. However, these gains are **highly sensitive to prompt design** and require careful alignment with pretraining formats, making them difficult to control and reproduce.
>
> * We therefore adopt **SMILES-only encoding** to isolate the variable of interest and ensure a stable and controlled comparison. Prompt-based approaches are complementary, but **not required to support our main claims**, and we leave them for future work.

---

> > ### Author Rebuttal · Reviewer_2NEK · 2026-04-02
> >
> > I thank the authors for their response and appreciate the inclusion of additional language models in the evaluation. I am not sure I agree that the two points raised are sufficient enough evidence that data leakage is not a concern in these settings (although I acknowledge this is not a criticism that is unique to just this paper). I maintain my original score of "weak accept".

---

> > > ### Author Response · Authors · 2026-04-05
> > >
> > > We thank the reviewer for confirming that the major concerns have been addressed and for acknowledging that potential data leakage is a broader issue in LLM-based methods. We are glad that the clarifications were helpful. We also sincerely appreciate the reviewer’s thoughtful feedback and careful engagement, which have helped improve the clarity and quality of the paper.
> > >
> > > Regarding the potential data leakage, we fully agree that this issue deserves careful scrutiny. To further assess it rigorously, we tracked the training data of all language and chemistry foundation models used in our study, and summarized them below.
> > >
> > > | Model     | Release year | Type                       | Training data                               |
> > > | --------- | ------------ | -------------------------- | ------------------------------------------- |
> > > | GPT-2     | 2019         | General LLM                | WebText                                     |
> > > | T5        | 2019         | General LLM                | C4 (Common Crawl)                           |
> > > | Llama-2   | 2023         | General LLM                | Public web data + licensed data             |
> > > | MolFormer | 2022         | Chemistry foundation model | ZINC + PubChem (~1.1B SMILES)               |
> > > | T5-Chem   | 2023         | Multi-task chem-text model | Pistachio reactions + procedures + ChEBI-20 |
> > >
> > >
> > > | Benchmark                                        | Year | Property                    | Label source                 |
> > > | ------------------------------------------------ | ---- | --------------------------- | ---------------------------- |
> > > | Redoxmers (Agarwal et al., 2021)                 | 2021 | Redox potential             | DFT simulation               |
> > > | Flow battery electrolytes (Agarwal et al., 2021) | 2021 | Solvation energy            | Physics-based simulation     |
> > > | Kinase inhibitors (Graff et al., 2021)           | 2021 | Docking score               | Docking simulator            |
> > > | Laser materials (Strieth-Kalthoff et al., 2024)  | 2024 | Oscillator strength         | TD-DFT simulation            |
> > > | Photovoltaics (Lopez et al., 2016)               | 2016 | Power conversion efficiency | Simulation / surrogate model |
> > > | Photoswitches (Griffiths et al., 2022)           | 2022 | Transition wavelength       | TD-DFT simulation            |
> > >
> > >
> > > The target properties across the six datasets are generated via physics-based simulations or task-specific pipelines (e.g., DFT, TD-DFT, molecular docking), defining a **pre-computed mapping from molecular structures to numerical values**.
> > > * **GPT-2** and **T5** were released in 2019, prior to most of the benchmark datasets used in this work. The only exception, the Photovoltaics dataset, was released via Figshare and is not part of the standard web-text corpora typically used for LLM pretraining.
> > > * **Llama-2** was trained on Public web data + licensed data, not on molecular property databases.
> > > * **T5-chem** was finetuned on T5 and the datasets used for training (e.g., Pistachio, ChEBI, procedures) contain structures, reactions, or textual descriptions, but **do not include computationally generated molecular properties**.
> > > * **MolFormer** is pretrained on large-scale molecular corpora such as ZINC and PubChem, which consist of SMILES strings **without associated property labels**. Importantly, these datasets do not include quantum chemistry outputs such as DFT calculations or docking scores.
> > >
> > > * Finally, our empirical results are consistent with this interpretation: if target values had been memorized during pretraining, we would expect strong direct predictive performance, which is not observed.
> > >
> > > While we cannot **fully rule out** the presence of isolated molecules in the some pretraining corpus used for large scale models, **systematic dataset-level leakage (i.e., access to structured molecule–property pairs) is highly unlikely**.  As we evaluate across multiple models and benchmarks, the consistent performance further suggests that data leakage is an unlikely explanation for the improved performance over non-LLM baselines.

---

### Decision · Program_Chairs · 2026-04-30

**Decision:**

Reject

**Comment:**

The paper presents a Bayesian optimization workflow which combines multiple moving components to construct tree-structured partitions of the search space with local classifier-based acquisition functions, candidate selection via Monte Carlo Tree Search and LLM-based clustering. As mentioned by Reviewer kiji, the method is too complex with many interacting components, making it hard to extract a single key contribution or assess where LLMs actually help. A comment was mentioned in the rebuttal that this work is primarily methodological  but the proposed approach is mostly a combination of likelihood free BO (Song et al) with LaMCTS (Wang et al). I want to emphasize that this is not a point for rejection in itself but then the combined approach should either show larger performance gains or improve our understanding. Unfortunately, the paper doesn't achieve that in its current form.  The presentation of the paper requires lot more work as well (as mentioned by Reviewers 2NEK and kiji).  The paper would benefit from incorporating comments by reviewers and focusing on demonstrating a clear focus and advantage and I request the authors to incorporate that in the future version. Overall, I recommend rejecting the paper and encourage the authors to revise and resubmit to an appropriate future venue by considering comments from the reviewers.